# MESSAGEix-Materials v1.1.0: Representation of Material Flows and Stocks in an Integrated Assessment Model

Gamze Ünlü[1,3], Florian Maczek[1], Jihoon Min[1], Stefan Frank [1], Fridolin Glatter[1], Paul Natsuo Kishimoto[1], Jan Streeck [3], Nina Eisenmenger [3], Dominik Wiedenhofer [3], Volker Krey[1,2]

[1]International Institute for Applied Systems Analysis, Laxenburg, 2361, Austria
[2] Industrial Ecology Programme and Energy Transitions Initiative, Norwegian University of Science and Technology (NTNU), Trondheim, 7491, Norway
[3] Institute of Social Ecology, University of Natural Resources and Life Sciences, Vienna (BOKU), Schottenfeldgasse 29, Vienna, 1070, Austria

*Correspondence to*: Gamze Ünlü (unlu@iiasa.ac.at)

**Abstract.** Extracting and processing raw materials into products in industry is a substantial source of $CO_2$ emissions, which is currently lacking process detail in many integrated assessment models (IAMs). To broaden the space of climate change mitigation options to materials-oriented strategies such as the circular economy and material efficiency measures in IAM scenario analysis, we develop the MESSAGEix-Materials module representing material flows and stocks within the MESSAGEix-GLOBIOM IAM framework. We provide a fully open-source model that can assess different industry decarbonization options under various climate targets for the most energy and emissions-intensive industries: Aluminum, iron and steel, cement, and petrochemicals. We illustrate the model's operation with a baseline and mitigation (2 degrees) scenario setup and validate base year results for 2020 against historical datasets. We also discuss the industry decarbonization pathways and material stocks of the electricity generation technologies resulting from the new model features. Next steps are to extend the model to other sectors, end-uses and materials, as well as the combined modelling of various supply- and demand-side measures.

## 1 Introduction

Extracting and processing raw materials into products which are used in various end-use sectors such as transportation, residential and commercial buildings, infrastructure, or consumer goods is a substantial source of $CO_2$ and greenhouse gases (GHG) emissions. Direct and indirect $CO_2$ emissions from industries constitute 35% of global GHG emissions (Lamb et al., 2021) and industry constitute 37% (157 EJ) of the global total final energy use in 2018 (IEA,2023c). Around 70% of the industrial $CO_2$ emissions and 50% of the final energy consumption are due to the production of bulk materials such as aluminum, cement, iron and steel, and petrochemicals (IEA, 2021c). Therefore, options on how to decarbonize these industries via energy- and material efficiency as well as demand-side measures are increasingly investigated (Watari et al., 2022, Lopez et al., 2023, Bhaskar et al., 2020). Understanding the potentials of these mitigation measures for reducing GHG emissions then requires model-based assessments of the entire life cycle of materials, from raw material extraction, industrial processing, to the use-phase of product stocks, as well as waste management, recycling and end-of-life treatment options.

Integrated Assessment Models (IAMs) have been used to generate scenarios for assessing energy and industry transformation for limiting climate change. However, recent reviews showed that many IAMs often lack the granularity, resolution, and framework to fully depict material flows and stocks (Bataille et al., 2021; Pauliuk et al., 2017; Stern, 2011). Many IAMs either omit physical material supply and demand, or model those in a simplified manner by directly relating material flows to economic indicators such as GDP (Pauliuk et al., 2017). Connecting material flows with end-use demand would however

enable modeling such cross-sectoral interactions among material stocks and flows, energy demand and GHG emissions, as well as various sectoral and service provisioning dynamics, which only very few models currently can do in a physically consistent and economy-wide manner (Wiedenhofer et al., 2024a). The tradition of partial equilibrium IAM modeling covers technology-rich models built on thermodynamic consistency for their respective sector such as the energy system. Only recently, partial equilibrium IAMs started to selectively represent material flows and stocks and their energy and emissions-

intensive production processes though the coverage is not the entire lifecycle of economy-wide materials. For example, Stegmann et al (2022) uses IMAGE to represent plastics, van Sluisveld et al (2021) to represent iron and steel, cement, chemicals, paper and pulp and Deetman et al (2021), (2020), (2018) for specific end-use sectors such as electricity, buildings, vehicles and appliances. Other partial equilibrium IAMs such as COFFE, POLES and PROMETHUS (Rochedo et al., 2016; Després et al., 2018; Fragkos et al., 2015) include iron and steel, cement, and chemicals.

Herein, we present MESSAGEix-Materials module, which aims to model the lifecycle of energy-intensive materials starting from raw material extraction, industrial processing, their accumulation as material stocks for various end-uses, to end-of-life waste flows and recycling within the MESSAGEix-GLOBIOM integrated assessment modeling framework (Krey et al., 2020). We develop the module based on a conceptual framework that integrates the traditions of energy systems modeling with

economy-wide material flow analysis. With this development, we provide a fully open-source model and its associated techno-economic data which can be used to assess different industry decarbonization options under various climate targets for the most energy- and emissions-intensive industries: iron and steel, cement, aluminum and petrochemicals.

MESSAGEix-Materials is operational within the global partial equilibrium MESSAGEix-GLOBIOM model which is based

on the MESSAGEix modeling framework (Huppmann et al., 2019) also incorporating macro-economic feedback using a stylized computable general equilibrium model, MACRO. The energy systems optimization model MESSAGE (Model for Energy Supply Strategy Alternatives and their General Environmental Impact) and the land-use model GLOBIOM (GLObal BIOsphere Model) (IIASA-IBF, 2023; Havlík et al., 2014) are the central components of the framework. Based on its scenarios GLOBIOM is linked to MESSAGE as a parametric land-use emulator (Fricko et al., 2017). MESSAGE is a linear programming

(LP) cost minimization energy engineering model with global coverage and perfect foresight as solving method. MESSAGEix-GLOBIOM provides a framework for representing a reference energy system with a full set of available energy conversion

technologies from resource extraction, imports and exports, conversion, transport, and distribution, to the provision of energy end-use services such as light, space conditioning, industrial production processes, and transportation. More details about the integration of MESSAGEix-Materials to MESSAGEix modeling framework are explained in Sect 2.1.


As a system engineering optimization model, MESSAGE*ix* is primarily used for medium- to long-term energy system planning, energy policy analysis, and scenario development (Huppmann et al., 2019; Messner and Strubegger, 1995). The model is designed to formulate and evaluate alternative energy supply strategies consonant with user-defined constraints such as limits on new investment, fuel availability, trade, environmental regulations and policies as well as diffusion rates of new

technologies (Gidden et al., 2023; Guo et al., 2022; Zhou et al., 2019). Environmental aspects can be analyzed by accounting, and if necessary, limiting the amounts of pollutants emitted by various technologies at various steps in energy supply. This helps to evaluate the impact of environmental regulations on energy system development. The principal results comprise, among others, estimates of technology-specific multi-sector response strategies for specific climate stabilization targets. By doing so, the model identifies the least-cost portfolio of mitigation technologies. The choice of individual mitigation options

across greenhouse gases and sectors is driven by the relative economics of the abatement measures, assuming full temporal and spatial flexibility (i.e., emissions-reduction measures are assumed to occur when and where they are cheapest to implement).

However so far, MESSAGEix-GLOBIOM only contains a simplified representation of the industry sector by distinguishing

three industrial energy demand categories: thermal, specific, and feedstock. Thermal demand, i.e., heat at different temperature levels, can be supplied by a variety of different energy carriers while specific demand requires electricity (or a decentralized technology to convert other energy carriers to electricity). So far, industrial production processes are not explicitly modeled for those energy demand types. Only the amount of cement production is linked to industrial thermal demand and the associated $CO_2$ emissions from the calcination process are accounted for explicitly (Krey et al., 2020). The current representation of

industrial energy demand is solely derived via GDP, which is not biophysically consistent as there is no link between physical resource extraction, material flows through sectors, accumulated material product stocks and the related energy use.

The MESSAGEix-Materials module presented herein advances this simplified industry sector representation and develops a consistent representation of material extraction, industrial production and processing technologies, together with the relevant

techno-economic data for the chosen industries. This enables explicitly representing the primary and secondary material flows occurring in industrial processes and, the resulting GHG emissions, opening the way for modeling materials-oriented climate change mitigation strategies. For the industries mentioned above, a set of low-carbon technologies in addition to the conventional technology options, are represented including recycling for metals. Because capital formation is a major driver of material demand and GHG emissions (Hertwich, 2021), we extend the MESSAGEix model formulation (Huppmann et al.,

2019) to also explicitly model material flows and stocks from the existing technologies in the model. We demonstrate the functionality of the new model formulation for the case of electricity generation technologies to allow endogenizing material demand and end-of-life materials from building up and transforming the infrastructure for these technologies. This formulation can also be used for other technologies in the model such as machinery in industry, or vehicles in the transportation sector, given data availability. One major challenge in developing a module that aims to have a comprehensive representation of the industry sector is gathering techno-economic data, especially with regional differentiation both for present and future. Table S1 in supplementary material lists all the data sources for different industries that were used in building up this module for 12 model regions. The details about model regions are seen in Table S2 in supplementary material. In that sense, we also provide a collection of techno-economic data sources with the open-source release of the module.

This version 1.1.0 of the MESSAGEix-Materials module is the first step of an ongoing development process to improve the economy-wide representation of material stocks and flows, industry sectors and their link to end-use demand. The overall aim is to represent the relations between societies' demand for service provisioning (e.g. building floorspace), the required material production and processing, as well as the associated energy and GHG emissions for industrial production and product stock operation ('stock-flow-service' nexus; Haberl et al., 2017; Wiedenhofer et al., 2024a).

In the remainder of this paper, Sect. 2 provides an overview of the MESSAGEix-Materials module including the modeling approach (Sect. 2.1), an integrative system definition (Sect. 2.2), the sector-specific representation for iron and steel, cement, aluminum, and petrochemicals (Sect. 2.3) and the demand side representation in the module (Sect. 2.4). Section 3 presents exemplary results from MESSAGEix-Materials while Sect. 4 discusses these results, limitations and next steps.

## 2 Model Description

### 2.1 Modeling Approach

"MESSAGEix-GLOBIOM" refers to a family of global- and country-scoped IAMs developed within the MESSAGEix framework. This framework comprises the MESSAGE linear program (LP) and related tools to operate the model. MESSAGE is a generic formulation of a least-cost optimization problem representing an abstract energy-economic-environmental system; it can (must) be parametrized to create models of any scope or resolution. The MESSAGEix-GLOBIOM model family parametrizes this generic LP with specific sets of technologies, spatial regions, time periods, and so forth. MESSAGEix is maintained and distributed as the open-source python package `message_ix` (https://docs.messageix.org/) with the core LP implemented in GAMS. This package also includes the MACRO computable general equilibrium (CGE) model, which can optionally be parametrized and solved iteratively with MESSAGE as "MESSAGE-MACRO" to represent demand responses to changing commodity prices. The `message_ix_models` (https://github.com/iiasa/message-ix-models,

) is another open-source python package that provides tools to (a) build, (b) solve, and (c) post-process or 'report' solution data for models in this family.

MESSAGEix-Materials is published as a module within this `message_ix_models` package. Using this module, any base model in the MESSAGEix-GLOBIOM family can be augmented with additional structural detail related to industry sectors, as described in Sect 2.3., and with data matching the resolution and parametrization of the base model. The MESSAGE LP is then solved with this added structure and data; then additional post-processing routines derive quantities of interest from the granular, full-resolution solution data. The same modular approach is used to derive MESSAGEix-GLOBIOM variants with enhanced detail in the water-energy-land nexus (Awais et al., 2023), buildings, and transport domains.

Figure 1 demonstrates the modeling approach and the workflow. Relevant techno-economic data of industry technologies are collected from various literature sources found in Table S1 in the supplementary material which are stored in either spreadsheet format or directly processed via python scripts. A MESSAGEix-GLOBIOM scenario is used as a base to build a scenario with the MESSAGEix-Materials module. One of the important advantages of this approach is that it enables easy integration with different MESSAGEix-GLOBIOM variants that have different spatial resolutions including country models (e.g., Orthofer et al., 2019) and different world regional resolutions covering 11, 12 or 14 regions. The example scenarios presented in Sect.3 use the 12-region global model. The materials module consists of scripts to process and prepare the model input data for all model parameters. In the final stage, the model is solved together with all the equations of the MESSAGE-GLOBIOM model in an integrated way. The energy system consists of all the extraction and fuel conversion technologies from primary to the final energy level. The new industry sector technologies that are added from the materials module use energy from the final energy level as input to produce material outputs. The reporting code produces a reporting output that is used for analysis. In addition, the integration of MESSAGEix-Materials to the current MESSAGEix-GLOBIOM can be seen in more detail in Fig. S1 in supplementary material.

Representing the material flows for the power sector technologies requires changes in the model formulation, because version 3.7 of message_ix doesn't endogenously consider the flow of material commodities linked to installing and retiring energy technology capacities during construction and retirement in the "COMMIDITY_BALANCE" equation (https://docs.messageix.org/en/stable/model/MESSAGE/model_core.html#auxiliary-commodity-balance). To endogenize material stocks and flows, the equation system of the MESSAGEix modeling framework had to be adjusted so that commodity flows are not only triggered by technology activities during the operational phase, but also by the construction, maintenance, and decommissioning of the technology capacity. This change in the equation system mostly affects the commodity balance equation and requires adding several parameters for the material demand and release intensities. Further description of the modification of the equations and the newly added parameters can be found in Fig. S2 in supplementary.

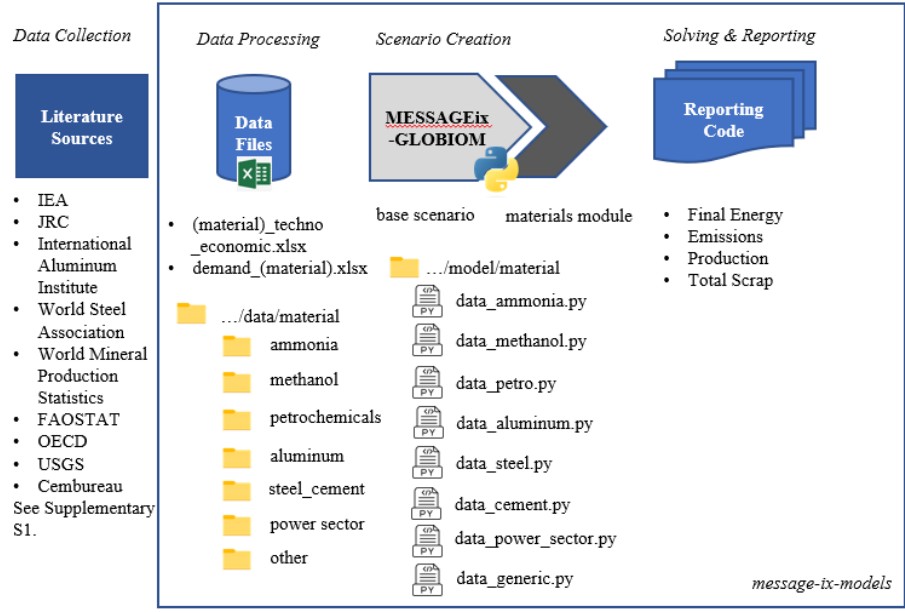

Figure 1: Workflow for using the materials module.

## 2.2 Integrative System Definition

In traditional energy systems modeling, the primary focus lies on the energy commodities that serve as inputs to various socio-economic processes and their implications for greenhouse gas emissions. Energy commodities are measured in energy units and traced from the extraction of primary energy carriers, along transformation processes to useful energy. Any energy losses along the supply-chain and in the use phase, e.g., waste heat, typically goes unreported as it dissipates, but could be reconstructed as residuals from the tracked energy flows. At the same time, flows of carbon and other GHGs are tracked in a thermodynamically correct, mass-balanced manner, because a key objective of energy systems models is to analyze GHG emission reduction strategies (Dodds et al., 2015; Herbst et al., 2012).

To address the mitigation potentials of materials-oriented strategies such as the circular economy and material efficiency, it becomes necessary to expand the scope of this energy-focused model towards fully covering material cycles and the dynamics of material stocks (Pauliuk et al., 2017). For this purpose, we draw on the field of industrial ecology, specifically Material Flow Analysis (MFA), which is widely used in the context of resource- and waste management (Graedel, 2019), as well as to quantify economy-wide resource use in accordance with system boundaries of the System of National and Environmental Accounts (Eisenmenger et al., 2020; Krausmann et al., 2017). MFA is based on the concept of social metabolism, conceptualizing society as socio-ecological 'organisms', requiring inputs of material and energy to build up, sustain and

reproduce their biophysical stocks of people, livestock and non-living material stocks, thereby producing waste and emissions (Fischer-Kowalski, 1998; Gerber and Scheidel, 2018). In recent years, dynamic MFA is increasingly used to implement a systemic, economy-wide perspective and to simulate stock/demand- or inflow/supply-driven scenarios, assessing technical/physical potentials and limits of material efficiency and circular economy strategies, e.g., lifetime extension, light weighting, reuse, recycling and downsizing (Hertwich et al., 2019; Worrell et al., 2016; Wiedenhofer et al., 2019; Lanau et al., 2019). The conservation of mass is the fundamental principle in MFA, which entails accounting for all by-products and waste flows occurring along material cycles from extraction to industrial processing, use as stocks, and end-of-life material flows (Graedel, 2019). This means that all material inputs into a system over a certain time period have to be equal to all outputs over the same period, plus/minus stock changes. A clear systems definition covering system boundaries, as well as processes and stocks and flows is indispensable for ensuring compliance with the mass-balance principle. A general material cycle system definition in economy-wide material flow analysis and additional information about the methodology can be seen in Fig. S3 in supplementary.

Combining energy systems modeling with economy-wide material flow analysis requires focusing on aspects which so far were not considered essential in traditional energy systems modeling. Therefore, we created a conceptual framework that outlines the targeted system boundaries of MESSAGEix-Materials within the physical earth system and the socio-economic system as commonly defined in the System of National and Environmental Accounts. In the conceptualization of the model and the definition of system boundaries, we draw on the latest literature and developments in economy-wide material and energy flow analysis (Plank et al., 2022; Wiedenhofer et al., 2024a; Pauliuk and Hertwich, 2015). Version 1.1.0 of MESSAGEix-Materials is a proof-of-concept implementation of the conceptual framework presented in Fig. 2. The figure shows the modeling of industrial and end-use processes ($P_{1,2,3, x}$) where material flows and stocks for cement, aluminum, steel and primary chemicals (ethylene, propylene, benzene, toluene, xylene) occur and are transformed from raw materials into products and waste. All the components seen in Figure 2 within the system boundary of MESSAGEix-Materials are endogenously implemented in the model but there are exceptions for some materials. These are explained at the end of this section. More details about the material flows, the mass balance equations and the corresponding material levels that are used to represent the flows in MESSAGEix-Materials in line with the energy systems modeling (see Sect. 2.3) are described in supplementary Table S3.

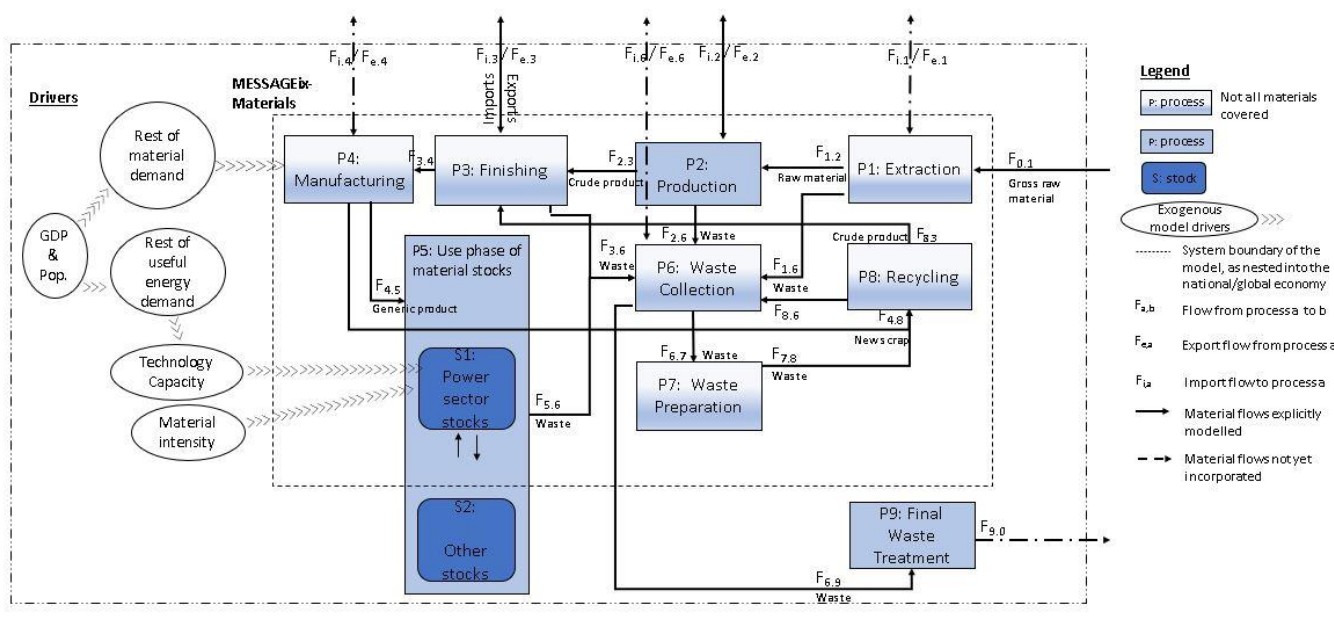

Figure 2: Generic representation of material flows and stocks in MESSAGEix-Materials.

In MESSAGEix-Materials, material demand for products drives industrial production of materials which accumulate as stocks, similar to the approach in other IAMs (Deetman et al., 2021). The drivers of the modelled demand are presented on the left-hand side of Fig.2. Overall, the combination of exogenous and endogenous demand determines the quantity of materials needed to produce material product stocks, which in turn determines raw material extraction quantities. The raw materials extraction process (P1) is defined in economy-wide material flow accounting principles (Eurostat, 2018; United Nations Environment

Programme, 2023) and it represents the process of extracting natural resources such as non-metallic minerals, ores, biomass, and fossil energy carriers from natural deposits, which are then further processed and traded (Plank et al., 2022). The model offers an example representation of detailed extraction processes through its implementation of the extraction of fossil fuels (https://docs.messageix.org/projects/global/en/latest/energy/resource/fossilfuel.html) such as coal, gas, and oil, which can be traded and are subsequently utilized as feedstocks in the chemical industry. Representation considers physical, technical and

economic resource availability, costs at which raw material can be brought to the surface and the losses and waste occurring during raw material extraction. Following the extraction phase, the raw materials move on to the production stage (P2), where detailed production technologies specific to each industry are explicitly represented. In this stage, the material and energy inputs related to the production process are provided together with production costs and emission factors. A more

comprehensive explanation of this phase can be found in Sect. 2.3 for each material. Subsequently, there's the Finishing (P3) and Manufacturing (P4) phase that produces generic products that then enter the use phase (P5).

In MESSAGEix-Materials, trade is represented for semi-finished or finished goods for steel, aluminum and chemicals, similarly to how energy commodities are traded in MESSAGEix-GLOBIOM. Each region can export and import through a global commodity pool without specifically tracking bilateral trade flows. Exporting requires "export capacity," which refers to the infrastructure and logistics capabilities of a region. This is modelled as a pseudo technology in MESSAGEix. Developing export capacity requires investment costs specific to each traded commodity, while the operational costs of shipping are treated as variable cost components. To determine the trade costs, we use data from World Bank and The United Nations Conference for Trade and Development (UNCTAD). Additionally, we introduce commodity-specific trade constraints and incorporate historical trade data (see Sect. 2.3) to ensure each region maintains a minimum degree of self-sufficiency and that the model aligns with current trade flows. We use trade statistics from selected data providers for each bulk material to match trade quantities in the base model year. The data sources used for trade can be found in Supplementary S1. Supply regions are determined based on production and trade costs. If it is cheaper to produce and export a commodity from one region than to produce it locally in another, the model will opt for export. The historical capacity and activity for production of a commodity are crucial in assessing a region's supply potential. The model includes the historical production and activity data, therefore has a representation of the already existing production capacity in the first model years. It also determines when the capacity needs to be retired and when the region needs to invest in new capacity which will have impact on the production costs. Additionally, if there is already an established export capacity between two regions derived by the historical trade data, no extra investment costs are incurred. Future trade projections will follow the same principle, focusing on cost minimization. A more detailed analysis of trade behaviour can be conducted in a study with regional focus by including various cost components, such as import tariffs.

During manufacturing (P4), a fixed percentage of new scrap is formed in case of metals. This type of scrap requires less preparation before recycling and has higher quality as it is the direct product of manufacturing. It is usually directly used within manufacturing without going through the market first, unlike old scrap which is formed at the end-of-life of metals. Drivers of the use phase (P5) are explained in more detail below. Once products reach the end of their useful lifetime, they are collected by the Waste Collection process (P6) and further distributed either to Waste Preparation (P7) and to Recycling (P8) or Final Waste Treatment (P9), where they are not recycled but landfilled or deposited elsewhere. For aluminum and steel, waste (also denoted as old scrap) is distinguished based on scrap quality (Nakamura et al., 2014), forming a scrap supply curve with three different quality levels 1/2/3 without specific end-use distinction. Level 1 is the highest quality of scrap and 3 the lowest. Different initial designs and final use conditions of a product determine the ease of recycling which is reflected in different scrap qualities in the model. Based on a simple supply curve logic, scrap qualities are available in different quantities, where

medium quality scrap (2) is available the most with 50% and high (1) and low quality (3) scrap with 25%. A minimum recycling rate can be specified for Waste Collection (P6) of steel and aluminium, either based on historical recycling rates or regulatory policies in different regions. The recycling rates in the model can be higher than the specified minimum depending on the economic attractiveness of the recycling options compared to primary material production. In addition, there is a maximum recycling rate imposed to represent limitations of recycling (e.g., contamination of steel by alloys). The formulation of this representation can be seen in Supplementary S7. During Waste Preparation (P7), the energy intensity and costs of the preparation for recycling increase, as the quality of scrap declines. Lowest quality scrap (3) requires more technologically advanced sorting and dilution methods to achieve the same quality of recycled metal as the ones produced from high and medium quality scrap (1-2). After preparation, scrap is sent to Recycling (P8) where it is turned into final materials to be used in the Finishing & Manufacturing (P3 and P4) again. During this process, recycling losses (F8.6) are also considered. It is assumed that the recycled materials have the same quality after the recycling process. All the old scrap that is collected in the waste collection stage (F7.8) is used in recycling assuming scrap availability and collection rate are the main bottlenecks of the recycling process. Final waste treatment such as landfilling, incineration or waste-to-energy are defined as being outside of the system boundaries of the current model version 1.1.0. To maintain mass balance, all material flows including waste in various stages (F1.6, F2.6, F3.6, F5.6, F8.6) can be tracked at each stage by adding relevant reporting variables to the reporting code.

The calculation of material stocks in the model version 1.1.0 differs between the power sector which is composed of electricity generation technologies, and other sectors, as outlined in Fig. 2 (P5 use phase covering S1 and S2). In the case of the power sector (S1), the capacity of electricity generation technologies in the cost-minimization problem is determined by electricity demand, which is driven by energy service demand which IS linked to exogenously given population and GDP dynamics, as well as the cost competitiveness of different fuel-to-energy routes. The required material stocks and their associated material flows are then endogenously determined by the modeling framework for the electricity generation technologies. For example, material stock accumulation of the power sector (S1) is determined by the multiplication of the newly built electricity generation technology capacities (e.g. solar panels, hydropower facilities, fossil fuelled power plants etc.) and the exogenous material intensities of those technologies (see Sect. 2.4.1). Further details on the modified model formulation to enable this calculation can be found in Sect. 2.1. Material flows not related to power sector stocks (S2) are determined exogenously in a highly aggregated manner by using GDP and population driven demand quantities for product stocks. These stocks then are calculated based on this flow and GDP correlation as described in Sect. 2.4.2. A portion of the manufactured products goes into use-phase (P5) and becomes waste, currently not based on lifetimes but based on ratios used in the model, while the remainder can be considered as stock. For a certain material, that waste ratio is determined by using what is reported in the statistics. This ratio is precisely calculated by dividing overall waste quantity (from both long- and short-lived products) to the

total production quantity in the base year (World Steel Association, 2020; IAI, 2020). This base year ratio is used in all years
in the model to determine the waste quantity.

Because version 1.1.0 of MESSAGEix-Materials is a proof-of-concept implementation of the novel conceptual framework described above, there are some deviations how the implementation of the general system definition shown in Fig. 2 is achieved for specific materials and industries which are discussed in Sect. 2.3. For the next versions, we aim to achieve a comprehensive,
economy-wide operationalization across all materials and end-uses. These deviations stem from the strategic decision to initially prioritize the representation of the most energy- and emissions-intensive processes first. These deviations are: First, for non-metallic minerals and metals, we do not fully cover the raw material extraction and mining phase (P1). Specifically, we exclude the detailed representation of energy use, physical waste and losses during raw material extraction of gross ores. The raw materials are simply assigned a price tag based on the current market prices for the model base year. This price is
discounted into the future with a 5% rate. For steel, we only use the quantity of the iron ore required to produce one unit of steel in the production process. Similarly, for aluminum, the model only accounts for bauxite extraction quantity as input to the system and represents the refining process. For cement, the quantity of limestone to produce one unit of clinker is included in the model as the main material input to cement production. However, other non-metallic minerals like clay, shale, or sand and gravel, that are mixed with cement to form concrete, are excluded. Secondly, the current approach only explicitly models
product lifetimes for electricity generation technologies, while all other material stock end-uses are determined based on simplified and aggregated approach as described in Sect. 2.4.2. It is important to note that, due to lack of data for the multitude of chemicals used nowadays, there is no product end-use detail and therefore no stocks are represented explicitly, because those chemicals are contained in various products. The only exception is nitrogen fertilizer, which is modelled explicitly, however it does not accumulate as stock but is purposefully dissipated to the environment. Thirdly, in the waste collection
(P6), preparation (P7) and recycling (P8) phases, only metals are modelled, while end-of-life flows of chemicals and cement is not represented as recycling/downcycling of cement currently occurs only at very low levels (Cao et al., 2017). Finally, in this version of the model, Final Waste Treatment (P9) technologies, costs and their mass-balances are not included.

Despite these derivations from the general system definition, MESSAGEix-Materials version 1.1.0 serves as the foundation
for future work incorporating further material flows and product stocks as needed for specific research questions related to a particular sector and/or material. This version of the model is therefore introduced as a proof-of-concept, which exemplifies how energy systems modeling and economy-wide material flow analysis can be integrated to represent material stocks and flows and their energy requirements and GHG emissions implications. As next steps, the extraction and mining phase (P1) can be represented in detail for those other materials similarly as for fossil fuels[1.] In addition, explicit dynamic stock-flow
implementations as for electricity generation technologies can be introduced to the model to increase the endogenous coverage of other material stocks such as buildings, infrastructure or vehicles. Regarding the end-of-life, it is possible to extend cement

and chemical flows to P6, P7 and P8 by following the same structure as other materials. For chemicals, plastics need to be represented as an explicit commodity in the model for a better representation of waste treatment options.

**2.3 Material Supply and Processing in Reference Energy and Material System**

MESSAGEix-Materials includes the explicit representation of technologies and processes from four key energy/emissions-intensive material industries: Steel, cement, aluminum, and petrochemicals. The materials are primarily chosen based on their substantial contributions to final energy use and emissions in the industry sector (Lamb et al., 2021). In addition, their end-use applications are considered for the potential to be combined with important demand side strategies such as the ones from

mobility, built-environment, machinery or packaging. The life cycle representation of different material industries in the model follows a generic structure as shown in the form of a *Reference Material System*, analogue to the common *Reference Energy System* representation used in energy system modeling (Beller, 1976) and is customized based on the process-specific differences between industries. A generic reference material system diagram for the model can be seen in supplementary Fig. S4. Different than Fig. 2, these series of figures that are used through Sect 2.3 provide information on explicitly how the

processes are modelled in MESSAGEix-Materials for a specific industry. Similar to the representation of energy commodities in energy engineering models such as MESSAGEix, depending on the stage of the material in its lifecycle, materials exist in different levels *primary_material, secondary_material, tertiary_material, final_material, useful_material, product, end_of_life, old_scrap 1/2/3*.

In Figures 3, 4 and 5, the use-phase and end-of-life phase of the resulting products are represented endogenously for the power sector while the rest is represented in the generic category 'Other_EOL'. The additional step 'Total_EOL' collects all the available waste at the total_end_of_life_1/2/3 level based on the scrap quality and availabilities a described in Sect. 2.2. From the waste available in these levels, the model decides how much to use via the scrap_recovery_1/2/3 technologies. Different energy inputs and costs are associated with the scrap preparation 1/2/3 technologies based on the respective scrap qualities.

Table S3 in supplementary shows how levels from reference material system diagrams are connected to the system boundaries and processes from Fig. 2.

This version of the model can be used as a basis to associate different scrap qualities with end-use sectors if it is linked to specific end-use demands and end-of-life flows considering lifetimes, such as for vehicles or buildings. For example, end-of-

life vehicles, machinery parts, and electronics are the highest sources of copper contamination in old scrap (low quality scrap), while new cars are the main end-use driving the demand for higher-quality steel (Nakamura et al., 2014).

Below the specific representation per industry sector is explained.

### 2.3.1 Iron and Steel

The iron and steel sector is one of the largest industrial GHG emitters among all bulk-material industries. Globally, the sector makes up 25% of the total direct industry $CO_2$ emissions with 2.1 Gt/year in 2018 (IEA, 2020c). Crude steel production in 2019 amounted to 1869 Mt/year (World Steel Association, 2020) and final energy use to 37 EJ/year in 2020, which makes steel production responsible for 24% of industrial final energy demand and 8% of global final energy demand (IEA, 2020c). Steel is used most notably in the construction sector to build buildings and infrastructure as well as to produce transportation

vehicles, machinery and appliances (Pauliuk et al., 2013). In addition, it will also be an important material for the energy transition, being used in low-carbon technologies such as solar panels, wind turbines, hydropower, and electric vehicles. Global steel demand has increased more than threefold since the 1970s as the rapid urbanization and buildup of infrastructure continues (IEA, 2020c). Steel is produced mainly through two routes. The more dominant practice is through *blast furnaces*, which produce pig iron from iron ore, and *basic oxygen furnaces*, which converts molten pig iron into steel by blowing pure

oxygen into the charge (BF-BOF process). There is a competing process called DRI-EAF routes (*direct reduction iron* and *electric arc furnace*), which goes through direct reduction of iron ore without melting it and uses electricity to smelt the charge and make raw steel. This process using EAF (not necessarily with DRI) has a lower energy use than BF-BOF which involves energy-intensive processing of raw materials. Also, EAF can be installed in smaller units for different market sizes and can be more economical than BF-BOF because it can fully rely on scrap metal as its input material with the possibility to vary the

input mix based on the market situation. While some of the countries with large steel-making capacities like the USA, India, and Italy rely heavily on the EAF steel-making process, still the BF-BOF is the dominating technology globally, and especially in China, which produces more than half of all steel supply globally. Specifically, China produces 90% of its steel from BF-BOF (World Steel Association, 2020).

The MESSAGEix-Materials model implements these two routes, while also considering old scrap recycling inputs into the material production process. The modeled process for the BF-BOF route includes the raw material preparation step, which involves coke oven and sinter/pellet plants. We also represent important low-carbon options in the steel industry in the model, such as CCS with top gas recirculation in blast furnaces, CCS in natural gas DRI furnace, 100% hydrogen use in DRI furnace, and replacing coke and coal with biomass in blast furnaces. Top gas recirculation with CCS in blast furnaces involves reusing

the gas that exits the furnace (top gas), which contains significant amounts of $CO_2$, CO, and $H_2$. Instead of releasing this gas, it is cleaned to remove some $CO_2$ and then recirculated back into the furnace. This process not only reduces emissions through carbon capture, but also increases energy efficiency by lowering coke consumption. Also, for the natural gas DRI technology, CCS can be added to capture $CO_2$ emissions. Typically, a DRI technology can operate with up to 30% hydrogen without modifications (MIDREX, 2022). To use 100% hydrogen in DRI technology, a retrofit option is added to the model. Finally,

we represent the option to replace coke in blast furnaces by biomass either partially or completely. The latter is particularly feasible in smaller mini blast furnaces, with capacities ranging from 0.06 to 0.4 Mt/year, commonly used in Latin America

(Fujihara et al., 2005). Figure 3 below shows the reference material system for iron and steel as represented in the model. In the figure, blast furnaces mode 1 is the traditional blast furnace, mode 2 is top gas recirculation (TGR) with CCS, while mode 3 and 4 represent the partial biomass substitution. In addition, a separate small size biomass blast furnace technology is represented that can fully operate with biomass. DRI technology has different modes with natural gas with or without CCS, as well as full hydrogen substitution. For electric arc furnace mode 1 and 3 represent primary production, operating with either coal or gas while mode 2 is the full scrap recycling mode.

Iron and steel was the world's 9[th] most traded product in 2021 in value (OEC, 2023). International trade of iron and steel is also an essential component to understand the technological changes and supply chain dynamics from raw materials to finished products. In the model, trade is represented at the useful_material level which delivers semi-finished steel after the finishing process. Regions can import and export semi-finished iron and steel from a global trade pool. For the calibration of trade, import and export numbers from "World Steel in Figures" report by World Steel Association are used (World Steel Association, 2020).

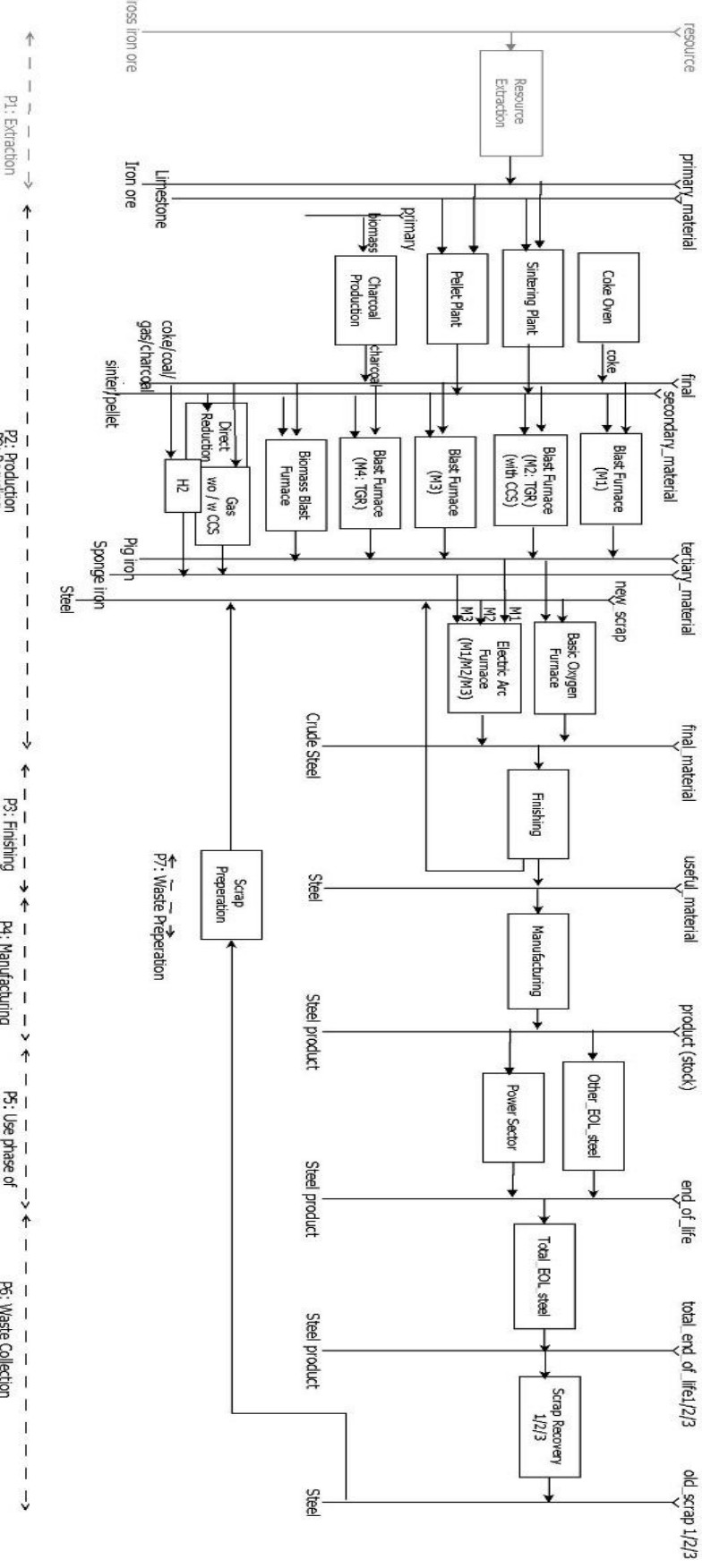

**Figure 3: Reference Material System for iron and steel.** Grey sections indicate those parts of economy-wide material cycles currently not represented in model version 1.1.0.

### 2.3.2 Aluminum

The aluminum industry mostly causes indirect emissions due to its high electricity demand of the industrial processes. The sector is directly responsible for almost 200 Mt/year of direct $CO_2$ emissions in 2018 (2.3% of the total industry emissions) and the number goes up to 1 Gt/year of $CO_2$ if the emissions from electricity consumption are included (IEA, 2023a). Production amounts to 100Mt/year in 2020 (IAI, 2020) and final energy use is around 5 EJ/year in 2020 (3% of the total industry final energy) (IEA, 2023b).

Aluminum is one of the non-ferrous metals that is widely used in the end-use sectors such as transportation, packaging, buildings, and consumer goods. The contemporary global aluminum stock in use has reached about 10% of that in known bauxite reserves and still no clear signs of saturation can yet be observed (Liu and Müller, 2013). In the context of the transformation to low carbon energy system, it plays an important role for strategies such as light weighting in the transport sector or for grid infrastructure expansion required to accommodate increasing renewable energy technologies (Kalt et al., 2021; Deetman et al., 2021).

The production process for aluminum consists of two main steps: refining and smelting. Refining is the step where the extracted bauxite mineral is converted to an intermediate material, alumina. The decarbonization potential mostly comes from smelting with its 78% share of energy consumption within the whole production process (IEA, 2009). In the model, we represent both refining and smelting steps. Refining takes bauxite as input from an unlimited supply with a relevant variable cost. The second process, smelting, is performed by two commercially available technology options; Prebake and Soderberg. Prebake and Soderberg technologies in aluminum production differ in anode manufacturing and usage. Prebake technology uses pre-baked, solid carbon anodes that are inserted into the electrolytic cells, offering higher energy efficiency due to better control over anode quality and cell conditions. In contrast, Soderberg technology involves continuously self-baking anodes from a paste added periodically, resulting in lower energy efficiency and higher operational costs. This technology is mostly used in locations where there is still existing capacity, with little new capacity additions. Both technologies require significant electricity input ranging from 13 to 17 kWh/kg which makes the primary energy intensity and indirect emissions very much dependent on the energy mix of a certain region. The liquid aluminum is then converted to the final product in the finishing and manufacturing processes.

The melting furnace and scrap preparation technologies enable the usage of old and new scrap as an alternative production path to smelting. Secondary production has around 10 times less energy requirement per ton of aluminium compared to primary production (Gautam et al., 2018). Figure 4 below shows the production processes for aluminum as represented in the model. The trade of aluminum is represented at the product level because finished aluminum from the manufacturing process is the

most traded form of aluminium (IAI, 2020). Regions can import and export finished aluminum at the product level from a global trade pool. To calibrate regional aluminum trade flows, we use data from the International Aluminum Institute, which reports material cycles from 1962 to 2021 (IAI, 2020).







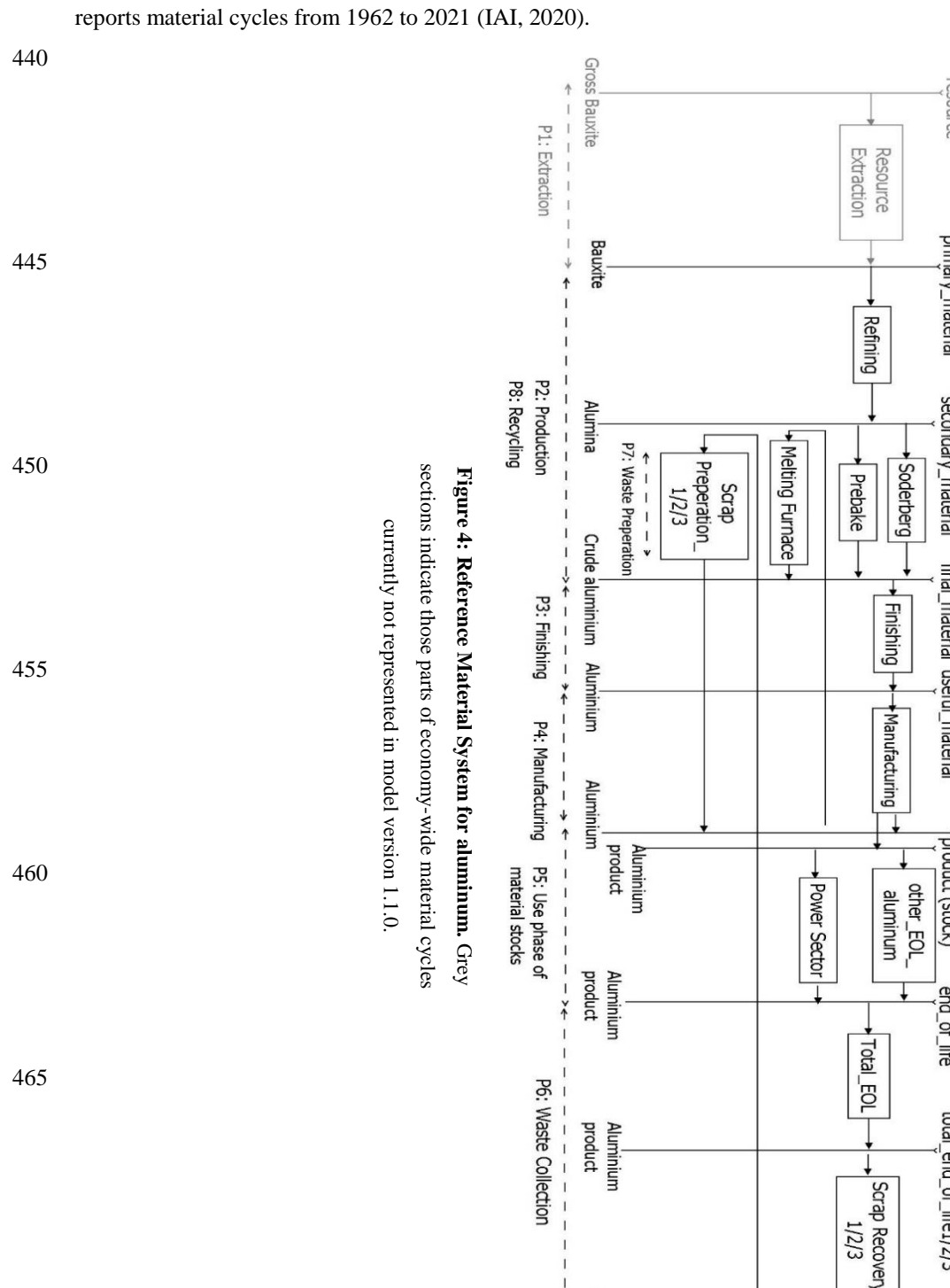

**Figure 4: Reference Material System for aluminum.** Grey sections indicate those parts of economy-wide material cycles currently not represented in model version 1.1.0.


### 2.3.3 Cement

While cement production consumes less energy than steelmaking, it is one of the most emission-intensive industries amounting to 2.3 Gt/year $CO_2$ in 2018, amounting to 27% of total industry emissions (IEA, 2020b). In 2020, cement production amounted to 4100 Mt/year, with a global thermal energy use intensity of 3.55 GJ/t (IEA, 2023d). Cement is widely used in buildings and
infrastructure development such as bridges or dams and due to ongoing urbanization and increasing affluence around the world, the demand for cement is expected to increase (Cao et al., 2017). About 60% of the emissions from cement production come out of a chemical process called calcination (Kermeli, 2016). Calcination is a process to remove carbon as carbon dioxide from limestone (calcium carbonate, $CaCO_3$) by heating it. This process happens in kilns, where raw material inputs (called 'raw meal') are heated to form clinker. Clinker is then ground with gypsum in a mill which becomes the end product, cement. Two
commercially available options to make clinker are included in the model: dry and wet processes. The wet process receives a wet mixture of washed raw materials, which consumes more energy to dry the materials. Two technologies are included in the model for the grinding process: ball mill, which is more conventional, and vertical mill, which is more energy-efficient for a higher cost. CCS options for the clinker-making stage are added to the model, given the importance of the technology for decarbonization of the industry. Figure 5 shows the reference material system for cement as represented in the model.


Cement is usually only traded at low levels compared to other commodities, mainly because of the high cost of road transport and the weight of the commodity, usually not making it worthwhile to transport over long distances. Global cement trading does account for 6-7% of production, most of which is transported by sea and usually used to balance out surpluses and shortages. Road deliveries of cement generally do not exceed distances of 150 km (European Commission, 2001b; Akram,
2013; Beirne & Kirchberger, 2021). Because MESSAGEix-Materials operates in 12 world regions, it is not possible to represent the local, intra-country or intra-region trade flows. Due to this reason and low overall trade volumes, cement trade is not represented in the model version 1.1.0.

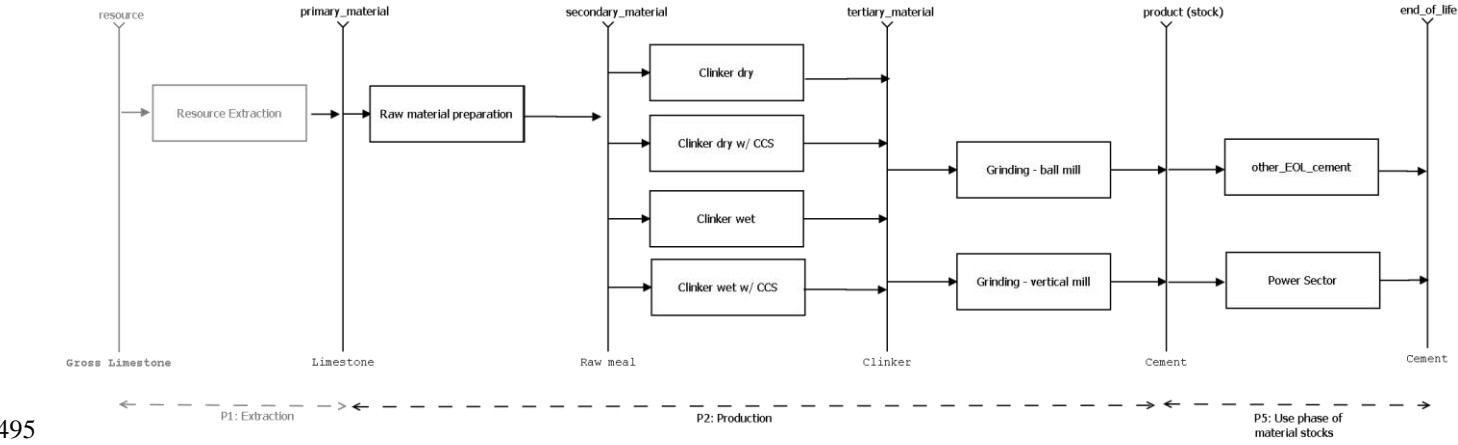


**Figure 5: Reference Material System for cement.**

Grey sections indicate those parts of economy-wide material cycles currently not represented in model version 1.1.0.

### 2.3.4 Petrochemicals


The chemicals and petrochemicals industry accounts for 14% of total industry $CO_2$ emissions, which amounted to 1.2 Gt/year $CO_2$ emissions in 2018 (IEA, 2020b). In 2020, final energy use, including feedstock use, reached 48 EJ/year, representing 30% of the industry's total final energy use (IEA, 2020a). Petrochemicals, a subset of chemicals derived from petroleum (oil) products such as naphtha or from natural gas such as ethane, are responsible for 90% of the total feedstock demand in chemical

production in 2018 (IEA, 2018). Almost half of the energy inputs to the sector are for feedstock use, which implies that there are less $CO_2$ emissions emitted from industrial processes compared to the steel and cement sectors, and that an important proportion of carbon remains in the final product.

Despite the complexity of the chemical sector, there are seven primary chemicals that provide the key inputs on which the bulk

of the chemical industry is based. These primary chemicals are ammonia, methanol and high value chemicals (ethylene, propylene and shortly known as BTX benzene, toluene, and mixed xylenes) which account for approximately two-thirds of the sector's total consumption of final energy products (IEA, 2018). In 2020, the production of these primary chemicals reached 543 Mt/year (IEA, 2021a; IEA, 2018; Methanol Institute, 2022).

In the model, carbon contained in the chemicals and plastics products is represented under the following assumptions. We differentiate products that are oxidized during use and long-lived non-oxidizing products drawing on the NEAT model, which differentiates chemical products based on their chemical stability during use (Neelis et al., 2005). In addition, we make use of the ratios of primary chemicals that end up in plastics and the share of the plastics that go to incineration. 85% of the high-

value chemicals and 65% of methanol are used in the production of plastics (Levi and Cullen, 2018). We use the plastic waste treatment projections of Geyer et al. (2017) to consider the $CO_2$ emissions from waste incineration. According to these projections, 28% of the plastics are incinerated in the base year 2020 and the incineration percentage is assumed to increase to 50% in 2050 (Geyer et al., 2017). The carbon that does not end up in plastics (released due to oxidation during use such as solvents), the carbon released because of incineration, and the carbon lost in steam cracking are released in the atmosphere in the emission accounting. The rest of the carbon is treated as stored within material product stocks and is therefore not accounted for in the chemical sector emissions.

**High-Value Chemicals**

The production of ethylene, propylene, and BTX, jointly referred to as high-value chemicals (HVCs) amounted to around 360 Mt/year in 2018 (IEA, 2018). HVCs are the main building blocks of plastics that are used in various end-uses ranging from packaging, consumer goods, to insulation of buildings. Refinery products are the most important feedstock to produce high-value chemicals. Therefore, the refinery representation is extended in the materials module to represent these intermediate products. Figure 6 shows the reference energy system for the extended refinery representation as represented in the model. Refineries can vary in terms of their structure and specific processes in different regions. In MESSAGEix-Materials, we represent a typical crude oil refinery reflecting the current operating refineries in North America based on PRELIM model (Abella et al., 2020). The processes that are represented include atmospheric and vacuum distillation, hydrotreating, catalytic cracking, catalytic reforming, coking, visbreaking and hydrocracking. The intermediate products of the refinery process are light and heavy fuel oil, naphtha, atmospheric and vacuum gasoil and residues, kerosene, diesel, gasoline, refinery gas and petroleum coke. These products can be used as feedstock in the chemicals industry, while the remainders are blended into two simple commodities light oil and fuel oil, based on their densities. Products that are not explicitly represented in the refinery are lubricants, bitumen (it is part of vacuum distillation residue but not a separate commodity) and paraffin waxes. *Pre_intermediate, desulfirized and intermediate* are levels added to represent the refinery products at their different stages.

Extending the refinery process enables detailed representation of the production of HVCs. Steam cracker is the main conventional technology that can use different feedstocks in different modes. In the model, the feedstock alternatives for steam cracker are two different types of gas oil (atmospheric and vacuum), naphtha, ethane, or propane. Each of these primary feedstocks results in a composition of HVCs with different ratios. Gas oils and naphtha are the products of the detailed refinery representation in the model while ethane and propane are formed by a natural gas processing plant. Production of ethylene from bioethanol is also included as a renewable production option. Bioethanol is supplied by ethanol synthesis via biomass gasification. The production pathways for HVCs as represented in the model are shown in Fig. 7. In addition, the methanol-to-olefins (MTO) process is represented which produces high- value chemicals which is explained in more detail below in the methanol section. Trade is represented for high-value chemicals as a single commodity at the useful_material level. For the

calibration of trade an enhanced data set (BACI) of UNSD Comtrade is used and the products that belong to HVC group are aggregated for MESSAGEix regions (see Supplementary S1). Also, a global cap is implemented for the trade volume of HVCs as maximum 20% of the production for all optimization years. This is because the feedstocks for HVCs are traded more widely than the HVCs themselves.


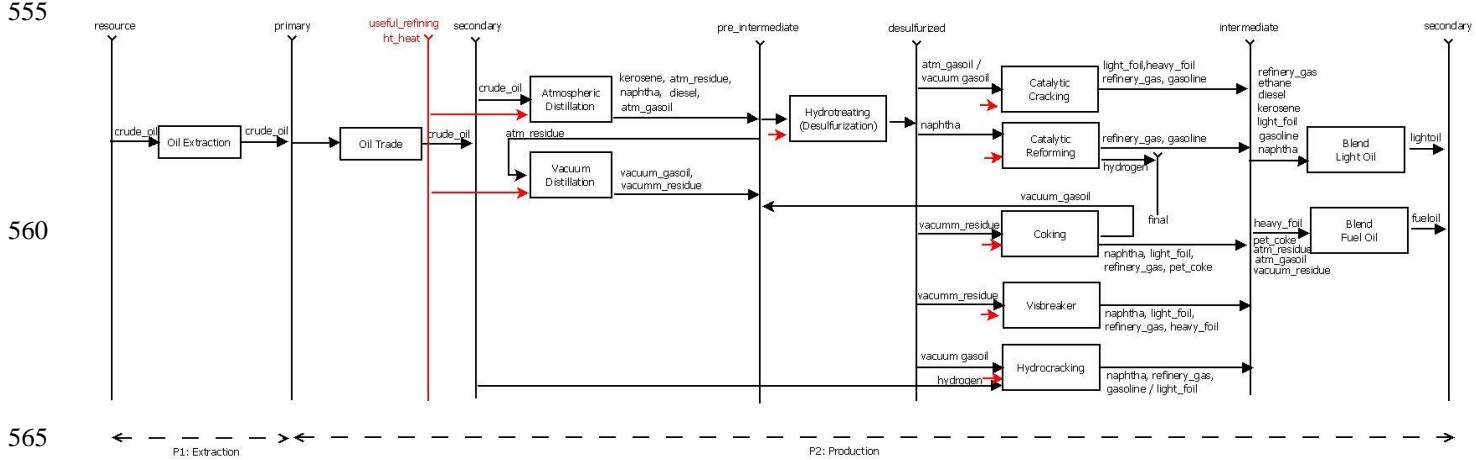



**Figure 6: Reference Energy System for refinery.**


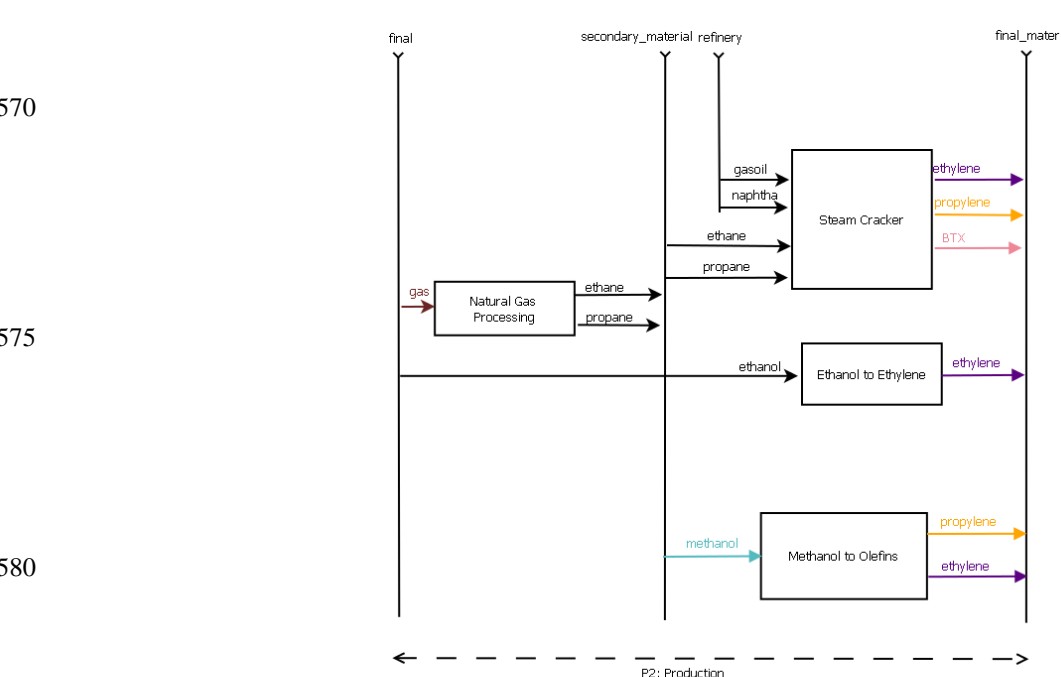



**Figure 7: Reference Material System for high-value chemicals.**

## Ammonia and Nitrogen Fertilizer

Global ammonia production for both fertiliser and industrial applications was 185 Mt/year in 2020. In the same year, production was completely fossil-fuel based, with 72% originating from natural gas, 26% from coal and the remainder from oil (IEA, 2021a). Ammonia is an input to all nitrogen fertilizer production processes. Among the nitrogen fertilizer products, urea-based fertilizer with high nitrogen content is widely used (58% of all N-based fertilizers in 2015) due to its high nutrient concentration and relatively low cost (Yara, 2018). The rest of the use cases for ammonia include cleaning products, refrigeration and air conditioning, production of plastics, textiles, explosives, food and beverage, and pharmaceutical industries. In addition, the use of ammonia as a fuel is promising as a carbon-free energy carrier produced via renewable sources which can be used in various applications, from power generation to transportation. However, there are various technical, safety and environmental challenges before it can be adopted widely. The potential end-use of ammonia as a fuel is currently not explicitly represented in the energy system model of MESSAGEix-GLOBIOM. Therefore, MESSAGEix-Materials version 1.1.0 only covers non-fuel use of ammonia.

MESSAGEix-Materials represents ammonia and N-fertilizer production processes currently with five different feedstock sources including coal, gas, biomass, fuel oil and hydrogen via electrolysis. In addition, the all technologies using carbonaceous feedstocks can be built with carbon capture and storage (CCS) technology. As we are not interested in the detailed material cycle of nitrogen, we do not represent specific types of N-fertilizers but treat them as a representative commodity type. For the trade implementation, currently MESSAGEix-Materials implements the trade of final N-fertilizer (about 30% of its production is traded globally) and the intermediate product ammonia (about 10% of global production is traded) (IEA, 2021a). The trade calibration has been conducted with identical methodology and data sources as for HVCs. In addition, GLOBIOM models the production, consumption, and the trade of the agricultural products. Figure 8 shows the reference material system for ammonia as represented in the model.

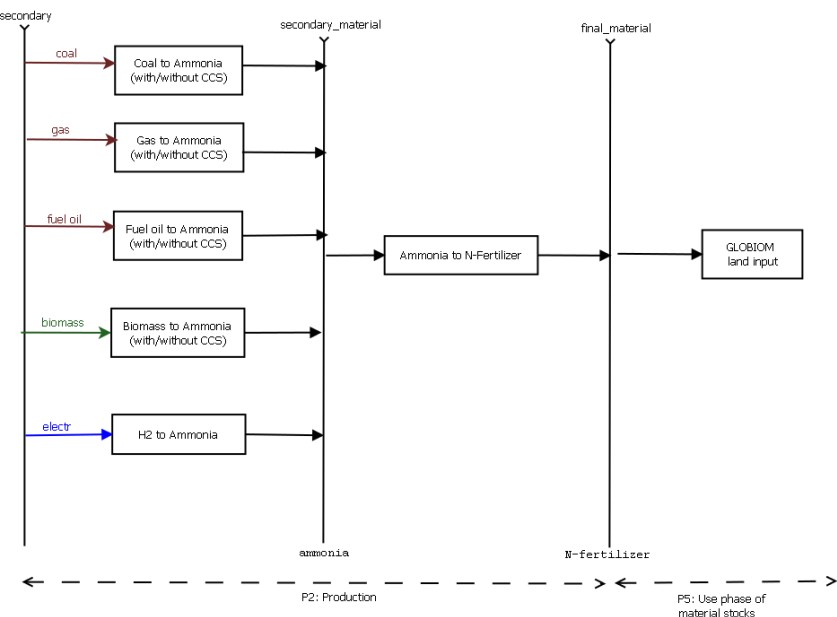

**Figure 8: Reference Material System for ammonia.**

**Methanol**

Methanol is the third major base chemical of the petrochemical sector with production just above 100 Mt/year in 2020 (Methanol Institute, 2022). Feedstock options are currently dominated by natural gas steam reforming and coal gasification. Coal gasification plants exist only in China, which produces half of the global methanol supply due to China's large methanol capacity. In MESSAGEix-Materials, various feedstock options are implemented including fossil (natural gas and coal) and low-carbon options with or without CCS. Low-carbon feedstocks are covered in the model with two technologies utilizing either biomass or syngas from hydrogen with captured carbon dioxide from CCS plants. In recent years, the production route "methanol-to-olefins" (MTO) has overtaken traditional chemical products as the single biggest methanol consumer. This technology is currently almost exclusively used in China, consuming around 25% of methanol production in China (Methanol Institute, 2022). Therefore, we include methanol-to-olefins technology in the model, which produces propylene and ethylene, featuring an alternative pathway to traditional oil-based petrochemistry. The second biggest share of methanol production is used to produce formaldehyde resins, which are mainly used in engineered wood products. The production of formaldehyde and the subsequent resin production is explicitly represented by individual technology instances. This enables endogenizing methanol demand coming from the construction industry for future applications of the model despite it is not included in this model version.

Besides traditional chemicals and MTO, a significant amount of total methanol production is used in various fuels. Besides direct fuel blending of methanol, the two major fuel related applications of methanol are methyl tert-butyl ether (MTBE) and

biodiesel production. MTBE is a gasoline additive that increases the combustion properties and is currently the most important substitute for lead as an anti-knocking agent. The second use case is biodiesel, mostly produced via the transesterification of

fatty acids with an alcohol. With methanol being the cheapest available alcohol, it is the most used alcohol for biodiesel production. The emerging methanol demand for these use cases is endogenously modelled in MESSAGEix-Materials by setting regional specific input shares for methanol in the MESSAGEix-GLOBIOM transport sector based on regional MTBE and biodiesel consumption. For an explicit representation in MESSAGEix-Materials, the methanol production technologies can operate in "fuel" or "feedstock" mode. The two modes are modelled with identical technical parameters and only serve

the purpose of being able to report methanol supply for energy and non-energy use separately. Figure 9 shows the reference material system for methanol as represented in the model.

Methanol is globally traded at high volumes, with South America and the Middle East being the major exporting regions at present. The trade calibration is done in the same manner as for HVCs and uses the same data sources.


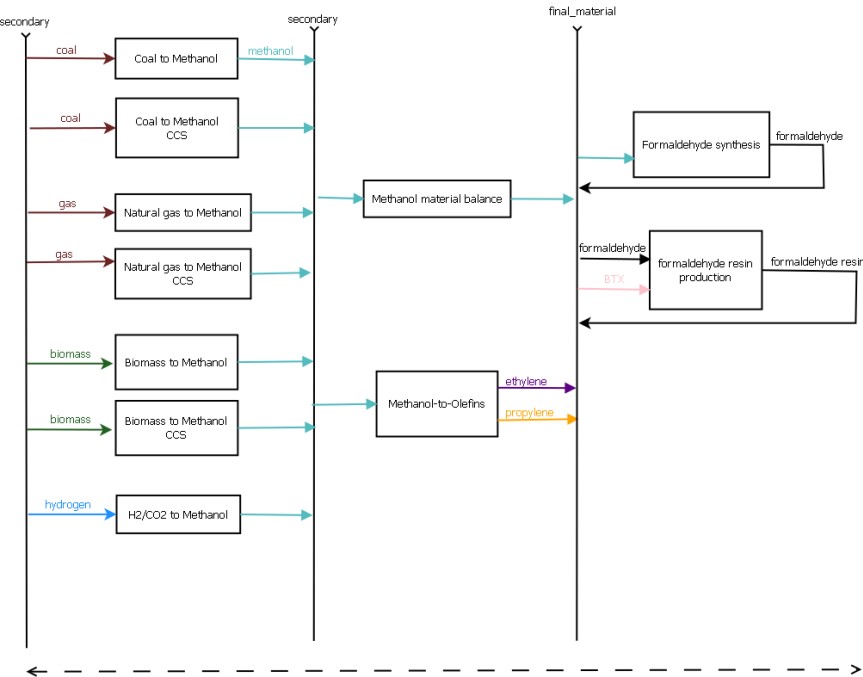

**Figure 9: Reference Material System for methanol.**

**2.4 Material Demand**

The demand for materials within MESSAGEix-Materials is determined in two different ways, depending on the end-use sectors. Section 2.4.1 explains the end-use sectors where material demand is endogenously generated by different parts of the model, such as the power sector's material stock demand, the demand for nitrogen fertilizer, and the explicitly modelled uses of methanol described in Sect. 2.3.4. For the other demand categories, the process of deriving material demands from GDP projections exogenously is elaborated in Sect. 2.4.2.


**2.4.1 Endogenous Material Demand**

Electrification is an important element of the transformation to a low-carbon energy system and many low-carbon electricity generation technologies, in particular those based on renewable energy sources, have in general higher material demand per

unit of installed capacity and in particular per unit of electricity generated than conventional thermal power plants (Arvesen et al., 2018). To incorporate this important linkage, we add material intensities for the most energy-intensive bulk materials used in power plant construction – steel, cement, and aluminum – to the MESSAGEix-Materials model. We rely on data from lifecycle analysis (LCA), specifically designed for the use within IAMs such as MESSAGEix-Materials (Arvesen et al., 2018; Kalt et al.,2021). Upon construction of power plants, a demand for the three bulk materials is generated endogenously based

on the material intensities in Arvesen et al (2018) and Kalt et al (2021) for hydropower, per generation technology, vintage and region. Power sector material stocks exhibit specific lifetimes, and upon the retirement of the capacity, end-of-life waste material is released, which is then collected and becomes available for recycling or goes into final waste management. Flows related to operation and maintenance are not included in the scenarios for this paper, though it is possible to investigate these flows as a result of the model developments described in Sect 2.1.


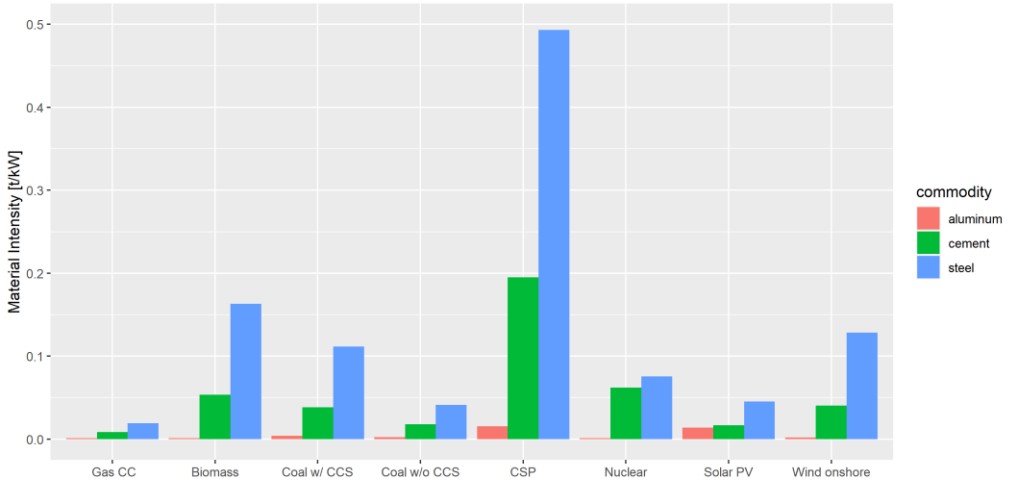

**Figure 10: Illustration of bulk material intensities for a subset of the electricity generation technologies in MESSAGEix-Materials. Data are shown for Western Europe in the year 2030. Source: 'mix' and 'residue' technology configurations from (Arvesen et al 2018).** Gas CC: Gas Combined Cycle, CSP: Concentrated Solar Power, Solar PV: Solar photovoltaics

Figure 10 provides an overview of material intensities (t/capacity of the power plant) for an illustrative subset of the electricity generation technologies in MESSAGEix-Materials. As can be seen, low-carbon energy technologies tend to have larger material intensities on average, than conventional generation technologies. However, two things should be noted in this context (Kalt et al., 2021), (i) there is a significant spread with material intensities varying by an order of magnitude among the groups of low-carbon and conventional generation technologies, and (ii) material intensities per unit of installed capacity need to be put into context of total possible energy generation, which depends on the operation time of the specific power plant. These ratios (material intensity / total operation time), also vary considerably among the technologies with illustrative full load hours ranging from 1000 hours per year for onshore wind or solar PV under less favourable conditions to potentially more than 8000 hours per year at which many nuclear power plants are operated. These factors are already taken into account in the energy system model MESSAGEix-GLOBIOM when deciding for a cost-effective electricity generation mix under different climate targets.

It is important to note that overall, in 2015, the material stocks of power plants are small, compared to total economy-wide material stocks (Kalt et al., 2021). Per material group, power plant stocks make up for 1-2% (aluminum and steel vs. all metals) and 1-3% (concrete) of the economy-wide material stocks estimated by Krausmann et al. (2018). However, it is still useful to endogenize these flows in the model for two reasons. Firstly, to investigate if the share of stocks undergoes a significant change with the increasing electrification under stringent climate policy. Secondly, as power sector technologies are already part of MESSAGEix-GLOBIOM, it's a good starting point to test new model functionalities for linking material flows and technology capacities. This same functionality which is described in Sect 2.1 more in details can be used to endogenize other material stocks that constitute a larger fraction. Data on end-use of stock-building material flows suggest that the majority of these materials are used in construction of buildings, infrastructure and other machinery (Liu and Müller, 2013; Pauliuk et al., 2013; Cao et al., 2017, Wiedenhofer at al., 2024b).

The second end-use sector in which the material demand is endogenously represented is nitrogen-fertilizer demand. For the N-fertilizer demand projections, we use the MESSAGEix-GLOBIOM emulator and import its synthetic fertilizer demand projections directly. Fertilizer demand in GLOBIOM is driven by future agricultural demand for food, feed, or other uses including bioenergy. The current implementation of the emulator includes a set of land-based climate mitigation scenarios, as well as a representation of selected land-use related SDGs such as moving towards low-meat diets and halving food waste, which all affect future nitrogen-fertilizer demand (Frank et al., 2021).

Finally, the methanol demand for the methanol-to-olefins route is linked to the exogenous high-value chemicals demand through the MTO technology. The methanol demand for fuel applications is driven by the demand for oil products in the transport sector, or the demand in industry as fuel based on the choice of the model.


### 2.4.2 Exogenous Material Demand

Regional demands for the total material demand for the three materials—steel, cement, and aluminum—are projected exogenously following the method suggested by van Ruijven et al (2016). This method chooses a best-fitting functional shape for the demand projection curve driven by per-capita GDP, which empirically is observed to have a saturating behaviour for each country. With the data extended with more observational years, we estimate updated curves for the three materials. We have found similarly to van Ruijven et al (2016) that a globally non-linear model (NLI) with an S-shaped relation between GDP per capita and material consumption fits best for the cement and aluminum in Eq. (1) and a variant in which per capita material demand is reduced over time as a result of efficiency improvement (NLIT) in Eq. (2) fits best for steel. The global consumption curve is used as a starting point and individual curves are derived for major steel-producing regions. The global curve for each material is calibrated to match the historical demand values for each region by modifying the maximum in the per capita consumption (PCC) curve (B) and per capita saturation level (a) parameters. In the formulations in Eq.1 and 2, T represents time and C per capita consumption.

Non-linear inverse (NLI): $C = ae^{\left(\frac{B}{GDP}\right)}$ (1)

Non-linear inverse with time-efficiency factor (NLIT): $C = ae^{\left(\frac{B}{GDP}\right)} * (1 - m)^{(T-2010)}$ (2)

This method is chosen as it can provide a first-order projection economy-wide material flows and allows us to fully cover the industry's energy demand resulting from those flows. However, the ultimate goal is to reduce the share of material demand that is currently exogenously derived. This can be achieved either directly in MESSAGEix-Materials, as was done in the power sector, or by integrating the model with other end-use sector models, for example, transportation and buildings by linking to MESSAGEix-Transport or MESSAGEix-Buildings (Mastrucci et al., 2021). The benefits of making more of the material demand endogenous are discussed in Section 4, Discussion and the Conclusion.

Due to lack of comprehensive historical data for petrochemicals demand, applying the same method which is explained above was not possible. Instead we rely on a methodology that uses IEA projections for high-value chemicals and the residual demand for methanol and ammonia (which can be used as fuel) that are not covered by endogenized demand drivers. The IEA projections are modelled by using regional GDP and historical demand intensity (IEA, Future of Petrochemicals Annex 2018).

However, the underlying data is not publicly available. Therefore, to account for differences in long term GDP trajectories, we
use the implicit income elasticity of the IEA demand projections and the GDP projections from the "Middle-of-the-road"
Shared Socio-economic Pathway (SSP2) (Dellink et al. 2017) to project the future chemicals demand in the model.

## 3 Model Results

### 3.1 Comparison of Base Year Results with Statistical Data

To validate the results, model values are compared with the reported statistical values in 2020 from different sources, mainly
IEA. More specific information on the sources for this comparison is listed in Table S4 in the supplementary material. Final
energy, $CO_2$ emissions, and production values calculated with MESSAGEix-Materials are compared to statistics for the year
2020 as shown in Table 1. For the model validation purposes we consider a deviation of +/-10% as acceptable and within
statistical uncertainties. For example, $CO_2$ emissions from fossil fuel combustion and industrial processes according to the
745 IPCC are known with an accuracy of about 8% (90% confidence interval; IPCC, 2023). In the below section, we explain the
reasoning behind some of the variations.

**Table 1: Comparison of MESSAGEix-Materials 2020 results with reported statistics.**

Numbers are rounded to the nearest integer

| | Aluminum | Iron&Steel | Chemicals | Cement |
|---|---|---|---|---|
| **Production (Mt/year)** | | | | |
| (MESSAGE 2020, Statistics) | (108, 100) | (2030, 1869) | (498,543) | (4187, 4187) |
| Abs Diff | +8 | +161 | -45 | 0 |
| Rel Dif | +8% | +8% | -8% | 0 |
| **Final Energy (EJ/year)** | | | | |
| (MESSAGE 2020, Statistics) | (6,6.2) | (31, 29) | (34, 37) | (15, 15) |
| Abs Diff | -0.2 | +2 | -3 | 0 |
| Rel Dif | -3% | +7% | -8% | 0 |
| **$CO_2$ Emissions (MtCO$_2$/year)** | | | | |
| (MESSAGE 2020, Statistics) | (216, 200) | (2458, 2654) | (924, 850) | (2506, 2435) |
| Abs Diff | 16 | -196 | +74 | +71 |
| Rel Diff | +8% | -7% | +9% | + 3% |

The relative differences in production levels (+/- 8% for aluminium, iron&steel and chemicals) can be explained as follows. For steel and aluminum, the material demand inputted into the model represents the final product, which includes losses from the manufacturing and finishing phases. However, the production figures reported by the model are based on crude material, which hasn't undergone further processing. As a result, there may be discrepancies between the model's output and statistical data, depending on the average loss rates used in the model. As for the chemicals, some of the demand is determined endogenously therefore there can be slight variations between the model values and numbers reported in the statistics. In the case of cement, the demand is provided exogenously to the model and the cement production exactly matches this number.

There are also variations in the levels of final energy use and $CO_2$ emissions. These differences are primarily due to variations in system boundary definitions. For the chemical industry, emissions and final energy statistics are collected from various sources for different chemicals. To compare these statistics with model results, we perform calculations to establish a system boundary that aligns with the model's framework. For example, the International Energy Agency (IEA) reports final energy use for all chemicals. We adjust this by multiplying their value with the share of primary chemicals in the final energy use to obtain a number that is comparable to our model values. A similar calculation is done for emissions to exclude the portion that result from the production of methanol that is used as fuel and not in chemical industry. These adjustments ensure our model results are within a reasonable range of emissions, typically with less than a +/-10% difference.

Iron and steel is one of the industries where accounting for the final energy and emissions is more complex. MESSAGEix emissions and final energy values include the coke and coal inputs to blast furnaces, which is partly converted to blast furnace gas. The compared IEA final energy values are calculated from the IEA energy balances to only include blast furnace and exclude coke oven energy consumption. $CO_2$ emissions are calculated by using the $CO_2$ intensity value reported by IEA which does not clearly state which part of the steel system boundary is included in the direct $CO_2$ intensity.

In addition, it is important to note that some statistics are not exactly for the year 2020 but instead for 2018 or 2019 such as aluminum emissions statistics which is reported for 2018 while the production value is from 2020. This can also cause some differences between the model and statistics values. Table S4 in supplementary shows more details on all these aspects.

**3.2 Comparison of Power Sector Material Stocks with the Literature**

Figure 11 compares the estimates of the material stocks in global power plants for MESSAGEix-Materials with the studies from Deetman et al. (2021) and Kalt et al. (2021). Over all generation technologies and materials, the estimates of total power plant material stocks are mostly within the same order of magnitude, but differ by up to factor 3 across studies (Fig. 11). This discrepancy is due to differences in the utilized values for two parameters: the installed generation capacities and the material intensities used for power plant technologies. Total installed capacity differs by 3-17% between studies (Table S5 in

supplementary). Per technology, differences range from virtually zero (for Solar CSP) up to 61% (for bioenergy & MSW).

Capacity differences in MESSAGEix compared to the other sources are a result of model calibration to electricity generation instead of generation capacity. The differences in capacities are small when compared to variations in assumed material intensities (Table S6 in supplementary). Across the five estimates in Fig. 11, material intensities per technology differ between factor 2 for cement in gas plants to factor 24 for cement in solar CSP plants. Large differences of over factor 10 are found for the material intensities of aluminum in gas, coal, nuclear and bioenergy & MSW plants. Overall, the largest absolute

differences manifest for aluminum stocks in Solar PV due to combination of capacity differences up to 27% and material intensity differences up to factor 6, as well as cement stocks in hydropower plants with capacity difference up to 13%, and material intensity differences up to factor 3. Despite the observed large differences in material intensities of plant technologies, the absolute estimates of total and almost all technology-specific material stocks from MESSAGEix-Materials are clearly within the range of the other two literature sources.


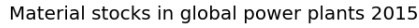

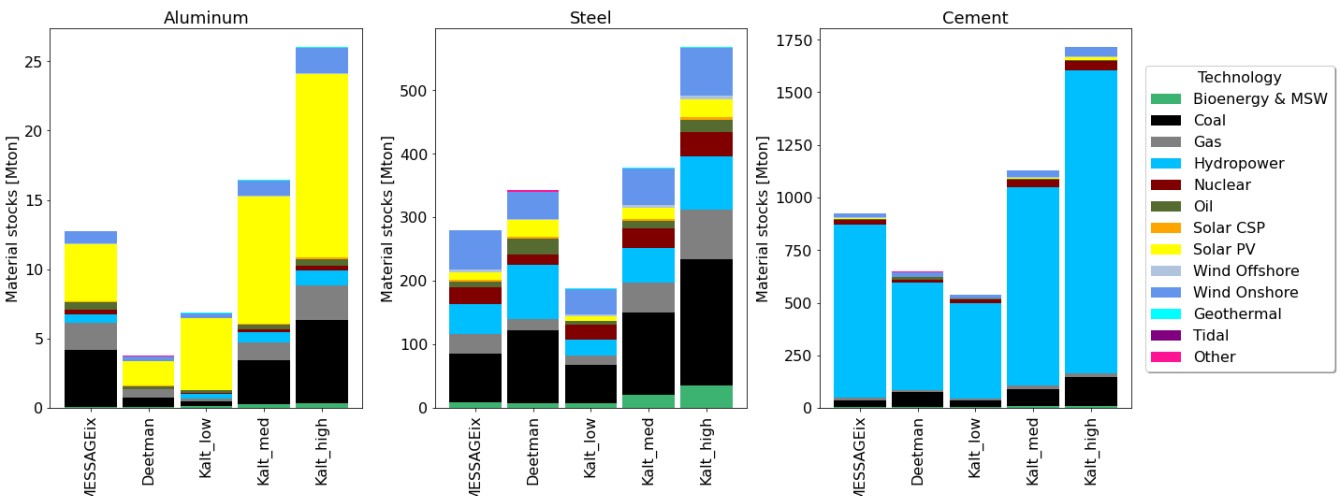

**Figure 11: Material stocks in global power generation technologies in 2015 based on this work (MESSAGEix-Materials), Deetman et al. (2021) and Kalt et al. (2021). The three estimates (low, med, high) by Kalt et al. (2021), result from using low, medium and high**

**material intensity assumption as derived from literature. The technology categories in the figure slightly differ from those of MESSAGEix-Materials due to mapping of all three data sources to a common technology set. Similarly, the comparison shows cement instead of concrete stocks. To convert concrete to cement stocks for Deetman et al. (2021) and Kalt et al. (2021) results, a cement content of 15% in concrete was assumed. MSW = municipal solid waste, CSP = concentrated solar power, PV = photovoltaic. For figure data please see supplementary Table S7.**

**3.3 Scenario Comparison**

The model results are compared for the two illustrative scenarios, 'NoPolicy' and '2 degrees' more specifically for the newly added industry sectors until the year 2070, the time at which total $CO_2$ emissions roughly reach net-zero. 'NoPolicy' is a baseline scenario without any additional policy constraints beyond developments until 2020 and thus serves as a counterfactual whereas the 2 degrees scenario aims at limiting global warming to 2 degrees by the end of the century. 2 degrees scenario uses global uniform carbon prices from a 1000 $GtCO_2$ "full century budget" scenario in line with 2 degrees (Riahi et al., 2021). A full century budget setup permits the budget to be temporarily overspent, as long as net-negative $CO_2$ emissions bring back cumulative $CO_2$ emissions to within the budget by 2100.

**3.3.1 Demand Side**

Currently, MESSAGEix-Materials derives the demand for materials either endogenously or exogenously as explained in Sect. 2.4. The following section provides results on material demand and stocks from the power sector as endogenously represented in MESSAGEix-Materials (see Sect. 2.4.1). Figure 12 shows the material stocks and technology capacities comparing a NoPolicy and a 2 degrees scenarios. Because of the increased electrification, the overall capacity of electricity generation technologies increases in 2 degrees scenario, specifically for low-carbon technologies such as wind and solar. The overall increase in capacity naturally implies an increase in the stocks of three bulk materials aluminum, steel, and cement. In 2070, the 2 degrees scenario has 60% more electricity generation capacity than the NoPolicy scenario (25 TW in NoPolicy, 40 TW in 2 degrees). Accordingly, the total material needs of electricity generation technologies in 2 degrees scenario is 2.2 times higher than the NoPolicy scenario (3049 Mt in NoPolicy, 6800 Mt in 2 degrees). Stocks of all three materials increase in similar amounts between the No Policy and 2 degrees scenario.

Producing the extra 3750 Mt of the bulk materials to build the electricity generation capacity of 40 TW in 2070 is equivalent to a 1.3 Gt $CO_2$ release considering the emissions intensity of the industrial sectors as in the 2 degrees scenario. The same increase in the material demand would be equivalent to 2.8 Gt $CO_2$ emissions in 2070 with the emission intensities of the NoPolicy scenario. To put these emissions into a scale, one can compare them with the emissions reduction from the end-use sectors between the two scenarios. $CO_2$ emissions from the demand side (transport, residential and commercial, industry) decrease by 20 Gt in 2070 in 2 degrees scenarios. Even though not all the decrease can be attributed to electrification, considering the increase from 37% to 51% in electrification on the demand side, we still can assume that the emission savings would be well enough to compensate for the increase resulting from the additional material demand. However, it should be noted that for a more complete picture of the material needs/stocks and for stock-flow consistency, all energy technologies should be considered including the replaced technologies by electrification (e.g., oil, coal-based heat providing technologies)

which might increase the material demand in NoPolicy. In addition, transmission and distribution infrastructure and storage technologies are important to consider with the increased electrification which can increase the material demand especially for

metals further in 2 degrees scenario (Kalt et al.,2021; Deetman et al.,2021). An exemplary investigation for electricity grids in North America with data from Kalt et al. (2021) and Deetman et al. (2021) showed that aluminum and steel in electricity grid infrastructure were 5-43 and 0.6-1.2 times larger than the material accumulated in power plants themselves (Streeck, 2022).

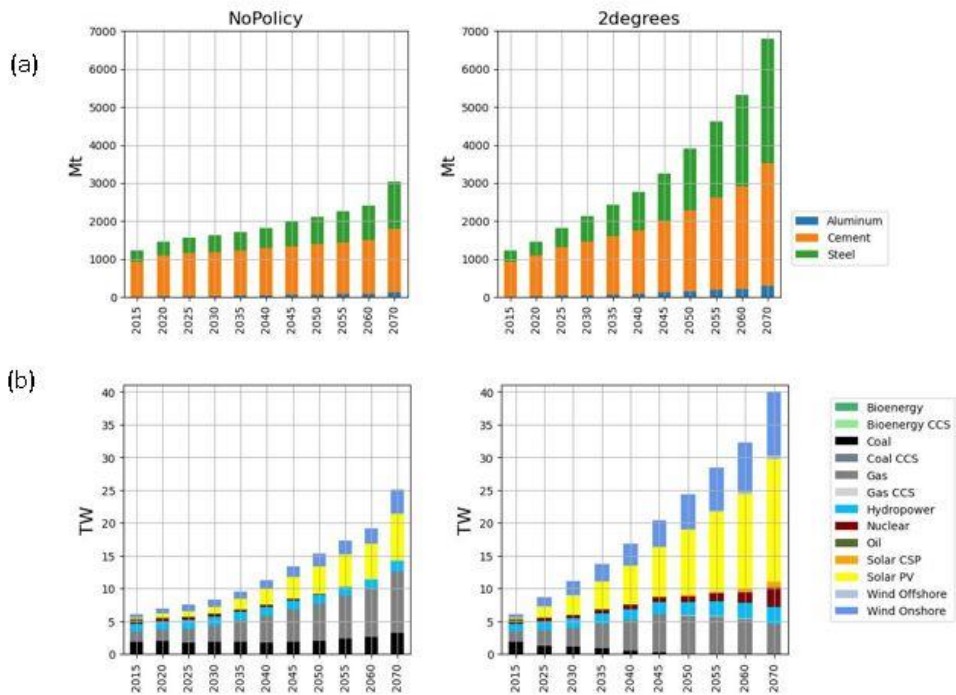

**Figure 12: (a) Power sector stocks by material, and (b) electricity generation capacities by technology.**

### 3.3.2 Supply Side

**Comparison of Materials and Non-Materials Versions**

As opposed to having one single exogenous energy demand for the industry, including explicit material flows in the model produces more technology-detailed pathways for industry decarbonization with different insights for each represented industry sector. Comparing the materials and non-materials version of the model (Fig. 13 & Table 2), we find that:

- The difference of emissions in 2020 results from the explicit representation of industrial production processes in materials module and different calibration approaches as a result of different model versions.

- Until 2050, the NoPolicy scenario without the materials module is growing slightly faster in terms of emissions. This contributes to higher emission reductions between No Policy and 2 degrees scenarios for non-material version. The factor behind this difference is that the two model versions have different representation of the industry energy demand. In non-materials version, industry useful energy demand is fully exogenous driven by GDP which results in higher values.

- Both the materials and non-materials versions exhibit a declining trend in the 2 degrees scenario, with the non-materials version generally showing a faster decrease until 2055.

- In Fig. 13, until 2025, the estimates for the 2 degrees scenario with materials module is stable, whereas without materials module emissions are declining. The limited potential for emissions reduction in the short term within the materials module stem from the more restrictive technology diffusion constraints imposed to specific industry technologies. More details on technology diffusion constraints can be found in S10 in the supplementary material.

- Finally, between 2060-2070, emissions from the 2 degrees scenario with the materials module are dropping much faster than without the materials module. This can be attributed to the increased utilization of CCS technologies in the chemicals, cement and steel industries, a feature absent in the non-materials version.

The explicit representation of industry technologies shows that the challenge of mitigating emissions from industry can vary at different times. As seen in this comparison, with the addition of the materials module, mitigation is more challenging until 2060, while it is faster after 2060 due to CCS technologies. The non-materials model uses a simplified representation by applying minimum and maximum share constraints on the industry's fuel mix without considering explicit technology costs. This approach results in higher emission reductions compared to the technology-rich representation of the materials module. This suggests that the simplification used in the non-materials model overestimates the industry's mitigation potential. It's essential to interpret this insight while acknowledging that the model does not encompass each available option within the industrial sectors. For instance, it does not account for price elasticities in material demands, and certain mitigation possibilities at the process level are not included in the analysis. In addition, because this comparison is only done for one scenario, it is important to note that the results can be affected by more stringent climate policies and different cost trajectories for the mitigation technologies.

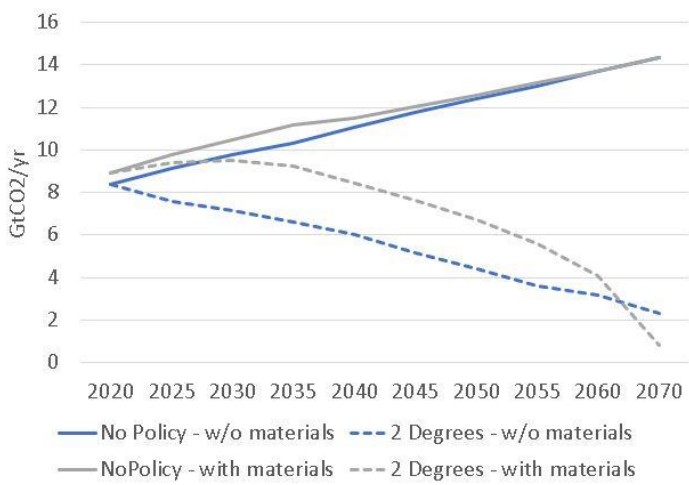

**Figure 13: Comparison of industrial emissions with and without the materials module in a No Policy and 2 degrees scenario.**

**Table 2: Emission reduction percentages in the 2 degrees scenarios with and without the materials module.**

| Emissions Change from NoPolicy to 2 degrees | 2020 | 2025 | 2030 | 2035 | 2040 | 2045 | 2050 | 2055 | 2060 | 2070 |
|---|---|---|---|---|---|---|---|---|---|---|
| Without Materials | 0% | -15% | -27% | -36% | -46% | -56% | -65% | -72% | -77% | -84% |
| With Materials | 0% | -4% | -9% | -17% | -27% | -37% | -47% | -57% | -70% | -94% |
| **Emissions Change of 2 degrees from year t-1 to t** | **2020** | **2025** | **2030** | **2035** | **2040** | **2045** | **2050** | **2055** | **2060** | **2070** |
| Without Materials | - | -4% | -5% | -8% | -9% | -15% | -15% | -18% | -12% | -28% |
| With Materials | - | 6% | 1% | -3% | -8% | -9% | -12% | -16% | -27% | -80% |

### Detailed Results from Materials Version

After comparing the materials and non-materials versions of the model, we now focus on the results from the materials model in detail. As a result of the climate policy, coal use in primary energy shows a substantial decrease in the 2 degrees scenario and instead is replaced by gas, wind, solar, and nuclear energy. Oil remains in the energy mix but to a lesser extent in the 2

degrees scenario. Looking at the $CO_2$ emissions, in the year 2020, the industry sector has a 44% share of the emissions followed by the transport sector with 40% while the emissions from Residential, Commercial & Agriculture Forestry, and Fishery (RC & AFOFI) is responsible for the remaining 16%. In the year 2070 in the NoPolicy scenario, the share of industry increases to 51%, transport is reduced to 35% and to 14% for RC&AFOFI. However, the 2 degrees scenario shows a substantial reduction of industrial emissions reducing its share of end-use emissions to 12%% in 2070 while transportation emissions rises to 76%

indicating that transport is much slower in decarbonization than the other end-use sectors. More details in primary energy by fuel, final energy by end-use, and $CO_2$ emissions by end-use can be seen in Supplementary Fig. S5.

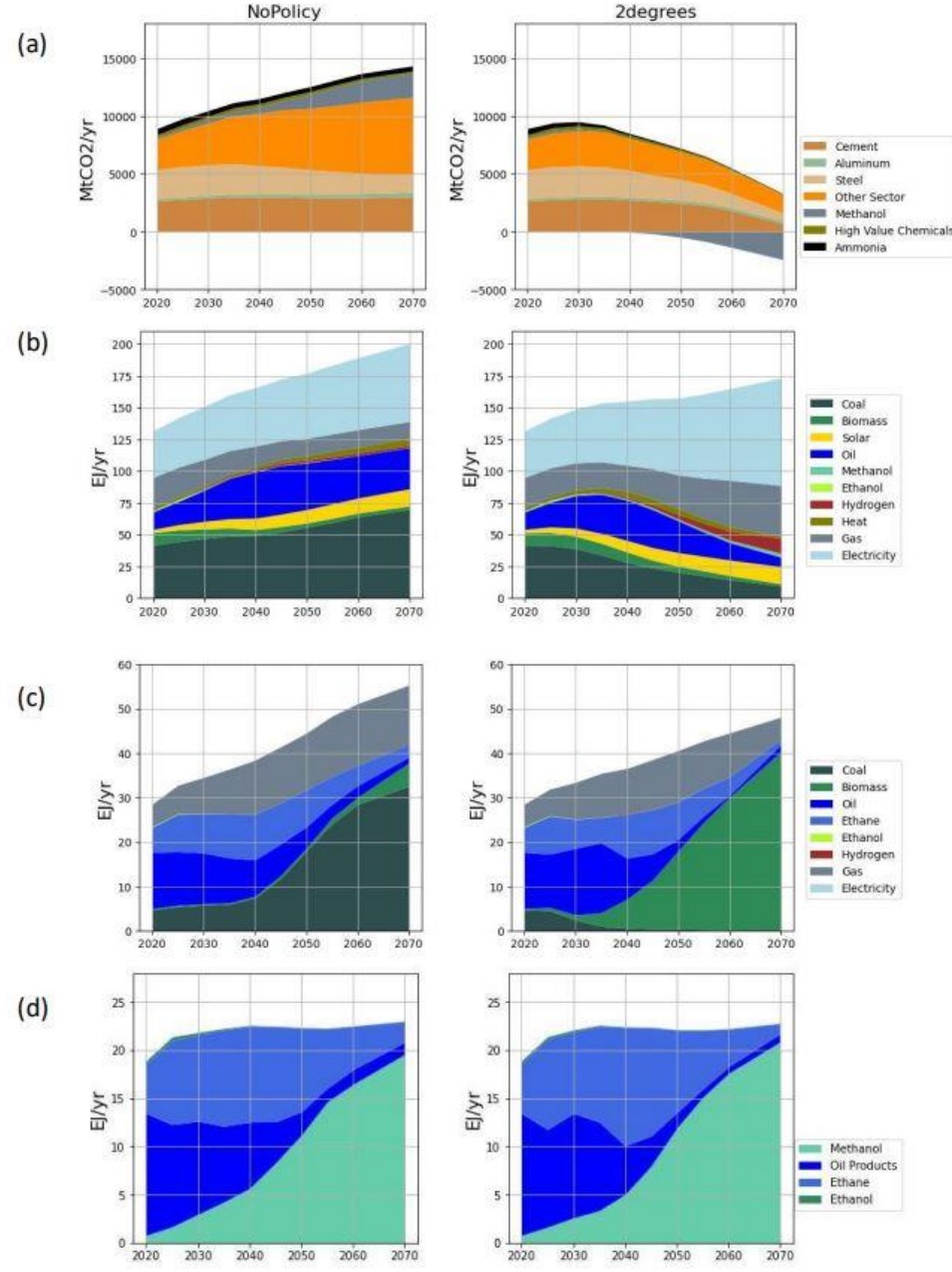

**Figure 14: CO₂ Emissions and Final Energy for Industry (a) Direct CO₂ Emissions by Industry Sectors (b) Final Energy Industry Excluding Non-Energy Use (c) Final Energy Non-Energy Use (d) High-Value Chemicals Feedstock Use.**

Figure 14 provides a more detailed look into the $CO_2$ emissions and final energy from the industry sectors that are represented in the model and the remaining "other" that is not explicitly covered. Other industry includes industries such as equipment and machinery, food, beverages and tobacco, textiles and construction. In panel a, in the year 2070, as one of the industries with a significant share in emissions, iron and steel sector contributes 6% to the industry emission reductions from No Policy to 2 degrees while a large chunk of the contribution comes from the chemicals and other sector with 38%. Cement follows this with an emission reduction share of 17% while the contribution of aluminum is minor (1%) as it has more indirect emissions coming from the production of the electricity rather than the production of aluminum. Indirect emissions from aluminum production in the model is 764 MtCO$_2$/year (7 t $CO_2$/t Al) in 2020 and becomes 796 Mt $CO_2$/year (4 t $CO_2$/t Al) in 2070 in No Policy scenario. In 2 degrees scenario, in 2070, this number goes down to 5 Mt $CO_2$/year (0.025 t $CO_2$/t Al) as the electricity mix becomes less carbon intensive.

The flexibility of sectors to reduce emissions depends on the availability of different cost-effective mitigation options. The industry with more cost-effective alternatives would be able to reduce its emissions more. In addition, MESSAGEix-Materials model incorporates technology diffusion constraints to realistically represent the inertia and dynamics of technology scaling up or down. The choice of parameterization affects both the pace and, in some cases, the type of technology adopted. In both NoPolicy and 2 degrees scenarios there are constraints for the scaling up and down of the technologies. Detailed information on technology diffusion constraints is provided in Supplementary S10, and parameter data can be accessed via the link in the Code Availability Section. However, in the case of the "other sector," which includes various smaller industries, the specific mitigation options for each are not yet explicitly represented. Therefore, the model makes fuel choices based on general minimum and maximum share constraints on fuels rather than detailed technological options. If technologies were represented in more detail, the potential for emission reduction could vary depending on the availability of low-cost mitigation options.

Panel b of Fig. 14 shows the transition of fuels used in the industry sectors from a NoPolicy to a 2 degrees climate scenario. Final Energy Use excluding non-energy (b) most notably indicates a significant decrease in coal (35%% to 6%) and an increase in gas (7% to 23%) and electricity (30% to 49%) shares from NoPolicy to 2 degrees in 2070. Hydrogen emerges as an alternative renewable source after 2050 from 0% to 8%, still with a limited share in 2070. The use of oil for high-temperature heat continues until 2040 in the aggregated other industry category in constant levels in both scenarios. Only after 2040, oil use starts decreasing in 2 degrees scenario.

**Chemicals Industry**

For non-energy use in Fig. 14 (c) there is not much change between the NoPolicy and 2 degrees scenarios in terms of oil and ethane use. Their use as feedstock diminishes after 2040 in both scenarios and is replaced by coal in NoPolicy and by biomass in 2 degrees scenario. In both cases, this is due to methanol becoming one of the main feedstocks to produce high-value

chemicals via the methanol-to-olefins (MTO) process. Panel d of Fig. 14 provides a more detailed look into the feedstock use in the production of high-value chemicals (HVC). The production routes do not change much between the scenarios but instead the source of methanol changes. In 2020, the feedstock of HVCs come from 30% ethane, 66% from oil, 3% from MTO and

1% from bioethanol. The bioethanol route does not scale up in the later years in 2 degrees scenario. In 2030, production relies on feedstocks with ethane accounting for 38%, oil for 50%, methanol for 11%, and bio-ethanol for less than 1%. However, by 2070, the feedstock composition shifts dramatically with methanol dominating at 92%, while ethane and oil both drop to 4%, and bio-ethanol remains at less than 1%. Final Energy graph for the Other Industry can also be seen in Supplementary Fig. S6.

The regional dynamics of MTO (can be seen in Fig. S7 in supplementary) shows that in 2020, almost the entire 3% of the capacity is installed in China. However, this distribution changes in the future years, and after 2040 there is much more regional diversification. In 2070, we see that Former Soviet Union (FSU), North America (NAM), and Middle East and North Africa (MEA) are the regions that deploy more capacity than others. In addition, Latin America and the Caribbean (LAM) deploys more MTO in 2 degrees scenario switching to biomass as a source of methanol.

Final Energy use for the two other primary chemicals, ammonia and methanol can be seen in Fig. 15. Fossil fuel use continues in 2 degrees scenario for ammonia production however combined with CCS, capturing around 350 $MtCO_2$ /year in 2070, as shown by the black line (panel a). For methanol production as mentioned earlier, the main feedstock switches from coal to biomass with CCS resulting in net negative emissions of close to 2.5 $GtCO_2$ in 2070 (panel b). Production via hydrogen is still

not cost-effective compared to CCS and is not deployed for both chemicals in 2 degrees scenario.

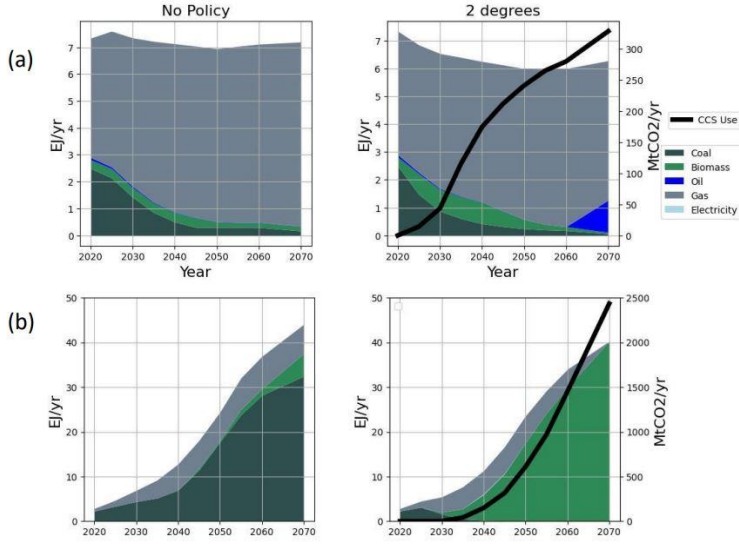

**Figure 15: Final Energy and CCS Use for (a) ammonia (b) methanol**.

**Iron and Steel Industry**

We find that the iron and steel industry is a major contributor to emissions and accounts for 6% of the total industry emission reductions projected in 2070. Figure 16 illustrates the technology mix within this sector, highlighting the changes over time. Currently, the blast furnace-basic oxygen furnace (BF-BOF) route, which uses coal to produce primary steel, dominates the industry. However, its share is projected to decrease from 63% in 2020 to 35% by 2070 under the No Policy scenario, and to 31% under the 2 Degrees scenario, with all remaining capacity equipped with carbon capture and storage (CCS) technology.

In the No Policy scenario, the BF-BOF route is already on a declining trend due to the lower levelized cost of production via the direct reduced iron-electric arc furnace (DRI-EAF) route, which utilizes gas or electricity, though the model does not fully account for the scalability and interchangeability of these two technological routes. As scrap availability increases over the years, the electric arc furnace (EAF) using scrap, which has lower energy intensity, gains a larger market share in the No Policy scenario. The availability of scrap is almost the same across all scenarios except the scrap release from power sector which 975 has a minor effect. The extent to which EAF technology is scaled up determines the amount of secondary production in each scenario.

In the 2 Degrees scenario, climate policies drive technological changes, particularly the adoption of carbon capture and storage (CCS) technologies. By 2070, CCS technologies are projected to capture 813 Mt/ year of $CO_2$ per year. Detailed capture 980 amounts over the years are available in Supplementary Fig. S8. Starting in 2035, the model begins deploying natural gas direct reduced iron (DRI) equipped with CCS. By 2050, all natural gas DRI capacity is expected to incorporate CCS. For blast furnaces, CCS is only implemented by 2055 for the remaining capacity, due to higher retrofit costs and a lower capture rate (65%) compared to natural gas DRI, which has a lower retrofit cost and a higher capture rate (85%). Data sources used for the technology parametrization are listed in Supplementary Table S1, and direct model input data is available via the link provided 985 in the Code Availability Section. Between 2035 and 2070, the share of electric arc furnace (EAF) using scrap in the 2 Degrees scenario increases reaching to 42% of the technology mix by 2070, compared to 36% in the No Policy scenario.

In the 2 Degrees scenario, we see that biomass and hydrogen do not play significant roles in reducing emissions in the iron and steel industry. The penetration of DRI-H2 technology is primarily driven by the cost of hydrogen production. Studies, 990 including those by Mission Impossible Partnership (2022) and Keramidas et al. (2024), project a low share of hydrogen when compared to other production routes. Keramidas et al. (2024) project that in a scenario dominated by CCS, hydrogen will only contribute 2-3% to the steel production, whereas an optimistic scenario with low-cost hydrogen electrolyzers and electricity could see a maximum of 15% hydrogen penetration by 2070.

To explore the conditions under which these technologies might be preferred, an additional scenario, where the steel sector must achieve net-zero emissions by 2070, is introduced. Unlike the 2 Degrees scenario, which applies only an economy-wide

climate policy, the Net Zero Steel scenario includes a sector-specific target for the steel industry alongside the broader climate goals. To achieve net-zero emissions in steel industry, we needed to relax the technology diffusion constraints after 2030. This change reveals that more radical and rapid technological shifts are necessary to meet the sectoral net-zero target. In Fig. 16, in Net Zero Steel scenario, although there is some inertia until 2030, the model shows a rapid transformation afterward, particularly with the phase-out of blast furnaces by 2035. This capacity is replaced by increased use of electric arc furnace (EAF) scrap, hydrogen, and natural gas DRI. In the following years, the natural gas DRI capacity is gradually retrofitted with carbon capture and storage (CCS), and the share of hydrogen in DRI production continues to grow. The contribution of hydrogen in steelmaking rises from 15% in 2035 to 48% by 2070. Natural gas DRI with CCS, used as a transitional technology, is phased out after 2060. By 2070, the technology mix comprises 49% EAF, 48% hydrogen, and 3% biomass. Biomass remains a minor component of the iron and steel industry due to its higher competitiveness and preference in the chemicals industry.

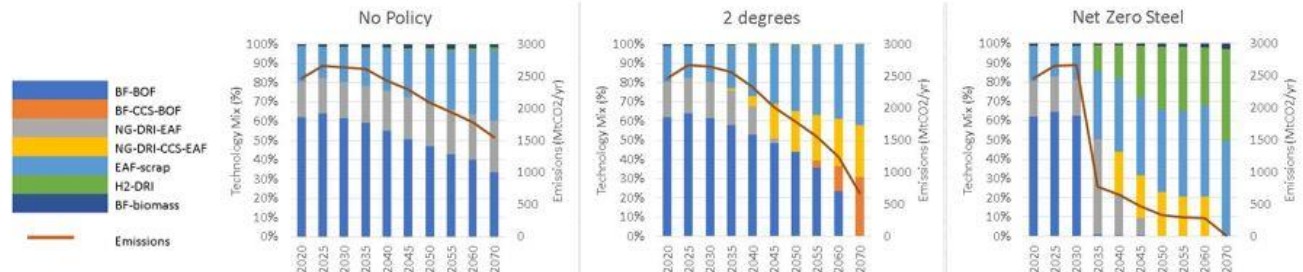

**Figure 16: Iron and steel industry technology mix and emissions**

Similar to our findings, Van Sluisveld et al. (2021) finds that for the iron and steel industry to reach a global net-zero emissions target by 2050, the industry must be 100% electrified, directly or indirectly, via increased EAF and hydrogen steel making. Otherwise, the industry will continue to produce residual emissions. Our Net Zero Steel scenario achieves full electrification by 2070, whereas 2 degrees scenario with more CCS preference still has residual emissions. Van Sluisveld et al. (2021) also notes that phasing out unabated blast furnaces could take until 2060, as seen in our 2-degree scenario. Unlike Van Sluisveld et al. (2021) which shows more radical short-term fuel switching to biomass, our analysis suggests that biofuels play a minor role, with CCS being the preferred strategy under a global economy-wide climate policy. Keramidas et al. (2024) further highlight that CCS developments in the latter half of the century are crucial for achieving net-zero emissions, projecting that the iron and steel sector could achieve net-zero as early as 2070.

**Cement**

Cement is another crucial industry that used coal to satisfy more than half of its energy needs in 2020. In the 2 degrees scenario coal use in cement production declines, and the use of gas and electricity increases until 2070 (Fig. 17) up to 20% and 46%

respectively. The electricity use increases particularly in the years 2055-2070 due to the rapid expansion of CCS and also increased use of electric kilns. The black line in Fig. 17 shows the process-related $CO_2$ emissions captured via CCS. The technology starts to scale up after 2050 and captures 1.4 Gt $CO_2$ in 2070. To satisfy the high-temperature heat demand oil is still used in the 2 degrees scenario with a 6% share of the energy demand in 2070. Methanol is also used as another fuel source with a 16% share in 2070 as well as a limited amount of hydrogen with a 4 % share.

Studies, including those by Van Sluisveld et al. (2021) and Müller et al. (2024), indicate that CCS is a crucial technology for decarbonizing the cement industry. According to Müller et al. (2024), in a scenario that complies with a 2°C temperature increase limit, the first CCS-equipped kilns will enter the market between 2025 and 2030, which is earlier than what our model projects. Both studies agree that fossil fuel use, such as oil, continues in kilns even under stringent climate policies. Additionally, Müller et al. (2024) maintain a significant share of coal use, whereas the MESSAGE model shows a sharp reduction. Both models predict increased use of natural gas. In Müller et al. (2024), electrification is lower, favoring biomass use in kilns instead. By 2060, they predict CCS will capture approximately 2.31 $GtCO_2$ /year, whereas our results suggest a much lower capture rate of 0.5 $GtCO_2$/year in 2060.

In our model results, compared to the chemicals and iron & steel industries, CCS technology scales up later in the cement industry. The main reason for this delayed adoption is the higher near-term costs of implementing CCS in cement production. However, a projected decrease in CCS costs for the cement industry in 2050 facilitates its adoption. Our scenarios indicate that in a system-wide cost-optimal scenario, earlier deployment of CCS in the cement industry can occur under stricter climate policies. Supplementary Fig. S9 and Fig. S10 illustrate an example scenario where CCS technology is adopted earlier in the cement sector due to very high carbon prices.

The timing of CCS implementation can also be explained by the differences in technological readiness. CCS technology is currently more advanced in the chemicals industry than in the cement industry (IEA, 2023e). In ammonia production, CCS is particularly suitable because the process naturally produces a high-concentration stream of $CO_2$, which can be captured efficiently. This captured $CO_2$ is used on-site to produce urea, and the remainder is prepared for transport or storage. Ammonia production with CCS is already operational in countries like Pakistan, Bahrain, the United States, and Norway, with a technological readiness level of 9 out of 13, indicating commercial viability (IEA, 2023e). The model shows CCS activity in ammonia production starting as early as 2025. Methanol production using biomass with CCS is less mature, with a readiness level of 8, indicating limited commercial applications today. In line with this, the model suggests that methanol production with CCS becomes more attractive around 2030 in a climate policy scenario. This technology is more attractive as it results in negative emissions, providing greater emissions reductions. Net emission reductions from CCS in the cement sector range from 72–90% due to energy requirements for $CO_2$ separation and upstream emissions (Viebahn et al., 2007). In the iron and

steel industry, natural gas direct reduced iron (DRI) technology has a readiness level of 9 and is operational in the United Arab Emirates. In contrast, CCS technology in the cement industry has a readiness level of 6-7, reflecting pre-commercial demonstration with no full-scale applications yet. The model accounts for these differences in technological availability, with CCS in ammonia available from 2020, while all other CCS technologies are set to become available starting in 2030.

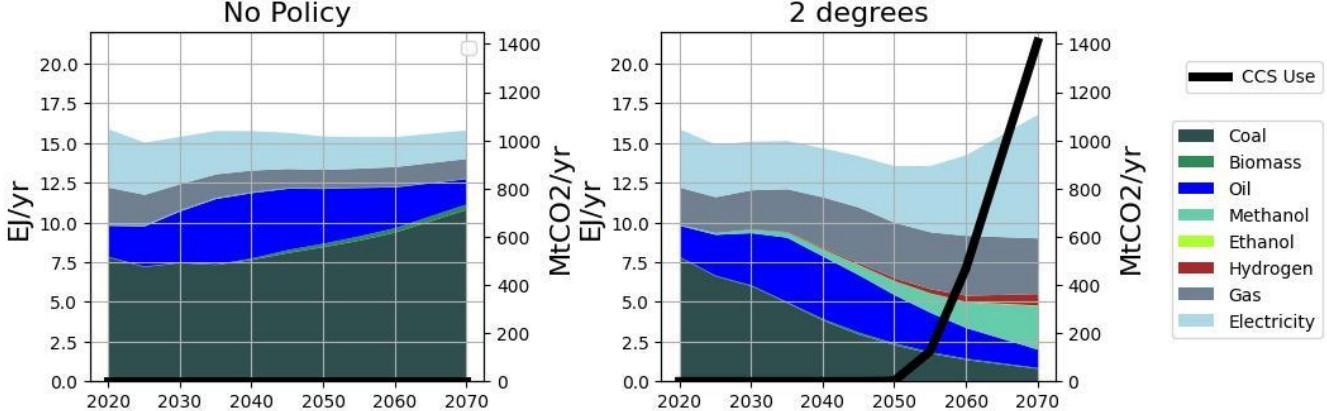

**Figure 17: Final Energy and CCS Use in cement industry.**

**4 Discussion and Conclusion**

4.1 Impact of Climate Policy on Industry Emissions

This simple scenario comparison exercise illustrates that the representation of the energy-materials nexus in integrated assessment models broadens the space of climate change mitigation options. We observe that incorporating explicit industry sector representation along with material stocks and flows can introduce both additional mitigation challenges and facilitate overcoming challenges, with effects varying across different time frames. By introducing additional industry sectors and mitigation options to the materials module, the impacts of mitigation challenges become more evident in contrast to a version without material considerations.

The comparison between the NoPolicy and 2 degrees scenarios reveals that climate policy can have a major effect on industry sector emissions which are reduced by 94% in 2070. With the MESSAGEix-Materials module, we opened the partial blackbox of the industry in conventional MESSAGEix enabling us to identify where the mitigation potentials of different industry sectors come from. Iron and steel as one of the major emissions-intensive industry offers emission reductions from NoPolicy to 2 degrees scenario with 6% of industry emission reductions in 2070 most of the reduction coming from increased EAF-scrap route and CCS technologies. As opposed to the economy-wide 2 degrees climate policy scenario, in an alternative scenario

where steel industry is pushed to reach net-zero in 2070, and where more rapid and radical technological changes are allowed, hydrogen steelmaking emerges as a crucial option, replacing the quickly phased-out blast furnaces. The cement industry as the next major emitter, has 17% of the industry emission reductions from NoPolicy to 2 degrees scenario in 2070 mostly due to the CCS use to reduce the process emissions and to some degree due to switching to cleaner fuels such as electricity, methanol and hydrogen to provide high-temperature heat for the processes. Chemical industry contributes to 38% of the emissions

reduction from NoPolicy to 2 degrees scenario. Our scenarios show that oil, ethane, and gas remain to be used as feedstock in the chemical industry while scaling up the methanol-to-olefins (MTO) route offers a bio-based alternative feedstock to produce high-value chemicals. Even though today MTO route is mainly used in China with coal as a feedstock, in future years regional diversification could increase and biomass with CCS has the potential to be used as the main source of methanol production. Another 38% of the emission reductions from NoPolicy to 2 degrees scenario originate from the "Other Industry" that is not

represented at the process level in the model which includes energy-intensive industries such as paper and pulp or glass as well as low-energy-intensive industries such as food and beverages, mining, and textiles.

4.2 Bridging Industrial Ecology and IAMs: Challenges and Benefits

Adding material-related dimensions into IAMs has its own challenges. IAMs primarily serve to examine the interplay between socioeconomics, climate, energy and land use systems in a quite aggregate manner. Conversely, industrial ecology tools, such as Material Flow Analysis (MFA) and Life Cycle Analysis (LCA), aim to trace and quantify material flows along with their environmental impacts and usually have more specific focus on certain products, sectors or materials. Consequently, it is challenging to bring together these two different scales. In addition, IAMs use a flow-oriented approach to e.g. represent energy

commodities and therefore, representing stock dynamics and comprehensively addressing the entire life cycle, especially the aspects pertaining to end-of-life and recycling, poses a challenge since these elements are typically not relevant for conventional energy system representations. On the other hand, IAMs offer a set of distinct advantages over traditional industrial ecology tools. While industrial ecology tools by now have lots of material flows and stock data, the "techno-economic" layer such as costs, production capacities, lifetimes of capital stocks are not as detailed as they are in IAMs. IAMs

incorporate techno-economics into decision-making processes, establishing feedback with the land-use system, and linking that to different energy carriers. In this regard, collecting the techno-economic data of material production technologies required by IAMs and the data on the regional differences for the technologies can be considered as another challenge to build up the materials module. Finally, one other benefit is related to the GDP driven material demand which does not consider biophysical material flows and stocks. This shortcoming can be overcome by a consistent modeling of the stock-flow-service

nexus that drives energy and material demand (Haberl et al., 2017; Wiedenhofer et al., 2024a). Material flows directly linked to service demand (e.g., passenger kilometer mobility demand, useful floor area of buildings) will offer a more accurate estimation of resource and material needs as providing services is the reason of the material needs of the society in the first

place. Furthermore, accounting for material product stock dynamics based on their lifetime will improve the estimation of end-of-life flows and the availability of recyclable materials. All these advantages can't be captured when a GDP based aggregate demand projection method is used.Version 1.1.0 of MESSAGEix-Materials serves as a proof-of-concept implementation for integrating material stocks and flows and holds potential for enhancement through collaborations between IAMs and industrial ecology tools.

4.3 Future Research Enabled by the New Modeling Advancements

MESSAGEix-Materials facilitates sharing data within the modeling community and enables collaborative work on improving the situation for techno-economic data in the industry sector by providing an open-source release of the model and the data. It also offers a methodological advancement for representing material stocks and flows within an integrated assessment model. The primary purpose of this advancement is to facilitate future research into the interactions between energy systems and materials as they are used in society to provide services. By examining these interactions, we can better understand and assess the potential of materials related strategies for climate change mitigation.

Future endogenization of economy-wide material demand and stocks could better represent the differences between the NoPolicy and 2 degrees scenarios in explicitly quantifying the additional material demand (and related energy use and emissions) not only for power generation technologies but also for grids, construction of buildings, infrastructure; transformations in the transport sector and related repercussions in industrial machinery. In line with this, endogenously connecting the material service demands (such as transportation or buildings) to the material production and end-of-life phase is crucial for better understanding the challenges of the transformation towards a sustainable net-zero socio-economic system. This is becoming increasingly relevant to analyse the synergies between materials-oriented strategies such as the circular economy, material efficiency, and other supply and demand side climate change mitigation options with high wellbeing for all (Creutzig et al., 2022; Sugiyama et al., 2024). In that sense representing the whole life cycle of the materials in the model enables IAMs to integrate circular economy measures and their links to carbon stocks and flows more accurately. Circular economy strategies such as recycling and use of renewable resources in materials production, can be directly represented in MESSAGEix-Materials as a result of the new model developments. Related to recycling aspect, one potential area of future work is to expand the model's representation of the end-of-life chemicals sector by incorporating plastics production and recycling processes, as well as fully representing waste treatment and landfills. This enables to investigate the trade-off between strategies like recycling vs. reducing the primary demand. As more material demand becomes endogenous or linked to demand-side models, additional strategies like increased reuse and repair can be modelled in the form of lifetime extension. One example for that is a household appliances model connected with MESSAEGix-Materials which can represent measures like sharing of appliances or increased repair. Similarly, demand-side sufficiency like reductions in service demand can also be

investigated with the explicit service provisioning indicators. For example, the MESSAGEix-Buildings model can produce a scenario where floor space per capita service demand saturates at lower levels, and this can be combined with the materials module to explore how these measures impact the energy system and climate goals (Mastrucci et al., 2021). Another potential use of the model is to investigate material substitution strategies such as calcined clays replacing clinkers which is an important mitigation option in cement industry (Scrivener et al., 2018). Including an explicit physical representation of materials improves the modeling of these strategies compared to previous model versions, which did not account for physical material flows.

These new developments also allow us also to investigate the material, energy, and emission implications of securing decent living standards (DLS) more consistently. DLS refer to a universal minimum set of basic goods and services required for a person to lead a healthy and fulfilling life (Narasimha & Min, 2018). The concept aims to define a threshold below which individuals lack the essential resources needed for well-being. To meet DLS, societies require resources which leads to material demand. With the MESSAGEix-Materials module it is possible to investigate the energy and climate impacts of the material requirements that arise from these minimum DLS (Virag et al., 2022; Vélez-Henao & Pauliuk, 2023).

Finally, as 38% of the modelled emission reductions come from "Other Industry", future work regarding the supply side includes adding further other important energy-intensive industries such as paper and pulp or glass. By that, emission reductions resulting from changes in these industries can be uncovered. In addition, extending the supply side with materials that play a strategic role in decarbonization such as copper, lithium, nickel, graphite or cobalt is important for having a comprehensive coverage of materials. Looking at the materials demand side, the power sector represented in MESSAGEix-Materials, currently only requires a small share of economy-wide material stocks. However, due to the increasing importance of electricity infrastructure and storage with progressing decarbonization of the energy and transport system, consideration of the material needs from electricity grids and storage technologies in MESSAGEix-Materials power stock estimates is a central future research agenda given the extensive energy system representation of IAMs. In sum, we expect this research area to continue to offer opportunities for further development and generate novel analysis and insights to complement more traditional mitigation options in the energy and land-use sectors.

**Authors Contributions**

GÜ, FM, JM, VK conceived the modeling framework. GÜ, FM, JM worked on model development with the coordination and supervision of VK. JS provided analysis on the comparison of power sector stocks between the literature and MESSAGEix-Materials. SF provided updated land-use scenarios from GLOBIOM model. FG and PNK provided information and guidance on releasing the model as open source in message-ix-models repository. DW and NE contributed to conceptual framing and the development of the integrative system definition in relation to material flow analysis perspective. GÜ led the manuscript framing and writing, preparation of results and conclusions. All authors reviewed and contributed to the manuscript writing.

## Code and Data Availability

Version 1.1.0 of MESSAGEix-Materials is available on the website: https://github.com/iiasa/message-ix-models/tree/update_steel_rebase under the Apache License, version 2.0. Model input data can be found in https://github.com/iiasa/message-ix-models/tree/update_steel_rebase/message_ix_models/data/material stored in xlsx format in separate folders for each industry. The same model version (1.1.0) that is used to produce the results in this paper is archived on Zenodo (https://doi.org/10.5281/zenodo.13121819), as are input data, scripts to run the model, and to generate the graphs for model results.

## Acknowledgments

The development of the initial MESSAGEix-Materials module was supported by the ALPS project, funded by the Research Institute for Innovative Technologies for the Earth (RITE). The addition of ammonia production and the link to fertilizer demand was developed as part of the NAVIGATE project with funding from the European Union's Horizon 2020 research and innovation programme under grant agreement No 821124. Modeling carbon dioxide removal options in the petrochemicals sector was supported by the European Union's Horizon 2020 research and innovation programme under the European Research Council (ERC) Grant Agreement No. 951542-GENIE-ERC-2020-SyG, "GeoEngineering and NegatIve Emissions pathways in Europe" (GENIE). Open-source data preparation received support under the SPIPA I and II India projects, GIZ contracts No. 81255150 and 83433054. JS and DW acknowledges support from the European Research Council (ERC) under the European Union's Horizon 2020 research and innovation programme, Grant Agreement No. 741950 and the European Union's Horizon Europe programme, Grant Agreement No. 101056810 (CircEUlar). JS also acknowledges funding from the Austrian Academy of Sciences (ÖAW) for their support during the IIASA YSSP 2022.

## Competing Interests

The authors declare that they have no conflict of interest.

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
