# Peer review of "MESSAGEix-Materials v1.1.0: Representation of Material Flows and Stocks in an Integrated Assessment Model"

_EGUsphere, 2023_

## Author Comment (AC2)

We sincerely thank the editor and all reviewers for their valuable feedback, which we have used to improve the quality of our manuscript. Here is our responses to the provided comments:

**Referee #1:**

This manuscript brings significant added value in enriching an integrated assessment model with physical flows and additional technologies, thereby making the model more relevant and pertinent to address the multi-dimensional questions arising from the challenge of addressing climate change. This work attempts to bridge the worlds of IAMs and industrial ecology, which is commendable.

The methodology is sound and the data collection appears to follow the state of the art of the literature. It is a paper that is useful for an IAM modeller and useful for a MESSAGEix modeller specifically, and indeed it serves its purpose as model documentation and validation of a first version of this module, awaiting future modelling improvements. However, it offers relatively few insights to the average reader. The paper's impact could be significantly enhanced if it had the ambition to bring more policy-relevant insights in terms of prioritizing actions to ensure a timely and cost-efficient decarbonization of the materials-industry nexus. This could be done by including more scenarios (for example with different climate policy intensities or other key drivers) and by including more detail on the mix of mitigation options by material sector. For iron and steel, including DRI with hydrogen and CCS options would immensely increase the relevance of the results. This is being implemented in many IAMs and has become an essential part of steel decarbonization pathways. For example:

using MESSAGEix: https://doi.org/10.1016/j.jclepro.2022.130813

using IMACLIM: https://doi.org/10.1080/14693062.2023.2187750

using POLES: https://doi.org/10.1016/j.egycc.2023.100121

To make improvements on these suggested points related to policy relevance of the paper and technology richness in different industries, we extended the representation of the iron and steel sector in the model. As suggested by reviewer, we include hydrogen DRI and CCS as well as use of biomass in blast furnaces in the new model version. We update the results and conclusions in the light of the newly added iron and steel technologies. As the main purpose of this paper is to serve as a methodological model documentation, we choose to use a NoPolicy and 2 degrees scenarios to validate the model but not a more elaborate scenario set with different climate policies and drivers. Scenarios with different policy aspects will be the scope of future papers. However, we included a few scenario variations in this paper to back up the results in the iron and steel and cement sectors. These are a net zero steel sector target scenario and a scenario with a very high carbon price that pushes the model to its limits to observe the uptake of the CCS technology in cement industry.

For cement, the timing and rate of deployment of CCS is strikingly different from what is presented for chemicals. This requires some explanation.

To address the comments about the timing of the CCS in cement industry, we have made some improvements in the model parametrization. These include cost revision for cement CCS and decrease of costs over time from 2030 to 2050 for cement CCS technology based on http://www.fp7-advance.eu/?q=content/industrial-sector-cement-guideline. The electricity and

high temperature heat needs of cement CCS are also revised based on (https://www.sciencedirect.com/science/article/pii/S0263876217303003) and (https://www.sciencedirect.com/science/article/pii/S1876610209000150) as they were found to be on the higher end. In addition, a sensitivity run with an extremely high carbon price reveals the limits and upper bounds of the model in terms of the CCS deployment in cement industry. This is added to the supplementary material and to the results section. We conclude that the later uptake is a result of the techno-economic costs of the CCS technology in cement industry and the earlier uptake of CCS in chemicals industry aligns well with the technological readiness of the CCS in chemical industry. More explanation is added to the text on this issue in Section 3.3.2, Cement.

There is a lot more detail in the petrochemicals sector compared to the other materials, which is also where this paper offers the most innovative work in how industry is represented in IAMs. The chemical industry is the most detailed sector, and most of the mitigation comes from that sector: these two things are inter-related, as the modelling choices drive mitigation potential. This has implications on the mitigation costs and potentials of other industrial sectors as well. More detailed work in each sector could provide options not previously considered and change the narrative on how hard to abate industrial emissions are.

We extended the representation of the iron and steel sector in the model as explained above.

Overall, this manuscript succeeds in making its case about represent more precise and explicit decarbonization pathways thanks to more sectoral detail. However, the mitigation potential appears broadly similar to before the model improvements (-85% in 2070). This might change if some of the questions above are addressed in the modelling (especially CCS coherence between cement and chemicals, and low-carbon steel processes). Interestingly, the mitigation potential in the new modelling is lower even compared to the old modelling in 2050, this could also be discussed in the final section.

It is mentioned that the mitigation potential appears similar to before the model improvements in 2070. This changes after the addition of the new iron and steel technologies and there is 94% reduction in industry related emissions in 2070, higher than the non-materials model version. Non-materials version has more reduced industry emissions in some years and more discussion is added on this as suggested by reviewer to Section 3.3.2, Comparison of Materials and Non-Materials Versions.

Some additional points:
The rationale to choose to represent these materials is not clearly made explicit: is it on the size of direct/indirect CO2 emissions? on the energy use of production? on how their demand is expected to evolve in low-carbon scenarios? The mention of the direct CO2 emissions at the beginning of each sub-section in 2.2 points to this being the deciding factor, but it's not clear.

The materials are primarily chosen based on their contribution to final energy and emissions in the industry sector. In addition, their end-use applications are considered for the potential to be combined with important demand side strategies such as the ones from mobility, built-environment, machinery or packaging in future papers. This information is now added to the beginning of Section 2.3.

On this topic, why choose to model aluminium instead of some other metal like copper?

Aluminum was chosen as one of the most energy intensive industries next to chemicals, iron and steel and cement. Although the carbon footprint of primary aluminium varies largely depending on the source(s) of electricity used, the process is highly electro-intensive, requiring around 14 MWh per tonne of metal, about seven times more than copper smelting. Aluminium production requires around 40% more energy than copper ((CRU copper: https://www.crugroup.com/knowledge-and-insights/insights/2021/cru-explains-copper-aluminium-smelting-emissions/).). Copper is being included in another version of the model not due to energy and emission intensiveness but for concerns of availability in the case of increased EVs in mobility.

The choice to explicitly model materials flows related to power generation technologies needs to be justified more, especially since this sector does not make up for a large share of demand for any of the materials detailed. Starting from a sector that makes up a larger share of materials demand (namely, buildings) would have made more intuitive sense.

Currently, the power sector does not constitute a significant share of the materials used. Despite that, the first reason for examining this sector is to determine how much this situation changes under climate policy, where the demand for power generation increases. This can also be complemented with the material needs of transmission and storage. The second reason is more operational: power sector technologies are already integrated into MESSAGEix as part of the energy systems model and therefore is a good starting point for testing the initial implementation of the changed model formulation. This change in the formulation that allows considering the flow of material commodities linked to installing and retiring technology capacities can also be used to represent other stocks endogenously. Incorporating other sectors, like buildings, require extra structural additions to the model. This type of study is planned but for a paper that focuses more on scenario and policy analysis whereas for the initial implementation we prefer to use power sector as a test case. An explanation is added to Section 2.4.1 Endogenous Material Demand.

The effort to be open source is appreciated -- although I did not see a link to the code and data. As such, the manuscript does not present several of its quantitative assumptions (especially regarding techno-economic parameters for production processes).

The link to the GitHub repository that includes the code is provided in Code and Data Availability Section. The data sources for techno-economic parameters are presented in Supplementary 1. The data itself is not presented directly in the text as there are many technologies in the model with various parameters and it is difficult to fit all the data in the text. However, the model data is accessible in the open-source model version in excel format separately for each industry including all the inputs and costs. The specific link for this is also provided under the Code and Data Availability Section.

**More detailed comments follow:**
L168-172 description of P1 phase: as made clear only later in the manuscript, this has not been modelled for the materials, so these considerations are not needed here. There is no trade of raw material, so the assumption is that the raw material is present domestically to satisfy the needs for production: at what price?

The description first describes what is implemented in the model for each phase P1, P2 etc. Then discusses what is not yet included in the current version of the model but are expected to be extended in the later model version by using a similar approach to the current representation.

Extraction is modeled for the feedstock materials such as coal, oil, gas which are used in chemicals industry. The trade of these material is also represented in the model. For the other raw materials which are not as much as detailed as the chemical's feedstock, there is an exogenous average raw material price attached based on the current market for the base year. The price is discounted into the future with a rate of 5%.

L186 it's not clear how the scrap levels 1/2/3 are determined: are these qualities of scrap (in terms of purity of material, from complex alloys to pure metal) or classes of ease of recovery (from low to high cost of recovery)? Are they related to end-uses of the material, with particular equipment lifetimes? How are the Power Sector and Other_EOL steps related to Scrap Recovery 1/2/3 in Figures 2-3-4?

Scrap level 1/2/3 are qualities of scrap which also affects the ease of scrap preparation. Level 1 has the highest quality (e.g., least impurities, least copper contamination) therefore it is less costly and energy intensive to prepare this type of scrap before the recycling step whereas 3 is the lowest quality. This type of scrap requires more technologically advanced sorting and dilution methods to be able to be used as recycled metal in any application. We can connect this information with the reference material system diagram as follows: At the moment in the model power sector and other end-of-life products release scrap and this is accumulated in total_end_of_life1/2/3. It is assumed medium quality scrap (level 2) has the highest availability with 50% of total scrap and the rest of the availability is divided between highest and lowest quality scrap (1 and 3). After this the scrap is recovered from these levels if the model prefers to use it based on production costs. Different energy and costs are associated with the scrap preparation 1_2_3. This version of the model can be used as basis to associate the different scrap quality levels with end-use sector demands if model is connected to other end-use demands such as vehicles or buildings. For example, end-of-life vehicles, machinery parts followed by electronics are the highest copper contaminating old scrap sectors, while the new cars are the main end-use behind the demand for the higher quality steel. (https://pubs.acs.org/doi/10.1021/acs.est.7b00997, https://www.sciencedirect.com/topics/materials-science/copper-scrap). However, this is not done in this version. More clarification is added to the text on these points.

L212, L232 refer to the relevant section here for clarity (2.3.2)

Changed as suggested.

L271 surely there is a better source than Statista

Changed to World Steel Association.

L296 What are the drivers and cost components that are included for trade? (same question for aluminium, chemicals and N-fertilizer/ammonia)

A more detailed explanation about this is added. Cost components considered are shipping costs as in the form of variable costs and the cost related to building an export capacity which includes the necessary infrastructure and logistics resources. The drivers of the trade are the production and trade costs which are driven by different factors such as existing production capacity, fuel prices and historical trade activity. More detailed explanation is added to the text.

L314 does this mean energy demand for refining is not included at all in MESSAGEix? Including this step to expand the model coverage does not seem to be overly complex.

Refining step for aluminum is added to the model to extend the coverage.

L318 why were two technologies singled out? how do they differ?

These two technologies are the ones that are used commercially at the moment for aluminum production therefore they are included in the model. The difference between two technologies is added to the text.

L339 "There are" L341 "we also model": in some cases it is not clear whether the authors are describing real-world operations, or how they plan to represent them in an enhanced version of this model, or how they have actually modelled it for this work.

Clarified as suggested.

L364 toluene and mixed xylenes are not mentioned again in the manuscript, why is it necessary to list them here?

The model includes the chemicals demand and production processes for the primary chemicals: ammonia, methanol, ethylene, propylene and shortly known as BTX benzene, toluene, and mixed xylenes. Since BTX is also included in the model it is mentioned in the introduction here. The sum of ethylene, propylene and BTX are referred as high value chemicals in the rest of the paper. In the text, a clarification is added for classifying BTX under high value chemicals.

L371 what is the source and data for these rations and shares?

The source is (Levi and Cullen, 2018) mentioned at the end of the sentence.

L399 "one of the renewable production options": are there other options?

Changed as suggested.

L417 please provide more detail on how ammonia-as-a-fuel demand is calculated in the model.

This information is provided in Section 2.4.2 Exogenous Material Demand. The approach is the same as deriving the petrochemicals demand.

L423 how do the "hydrogen via electrolysis" and green hydrogen processes differ? Are there multiple hydrogen production pathways as input to ammonia production?

The sentence is changed.

L446 "to allow endogenization": I understand that this is not modelled (yet)? If so, it should be clarified.

Clarified as suggested.

L448 please provide more detail on methanol use in "various" fuels and how the fuel/feedstock production technologies differ.

More details added as suggested.

L485 Presumably, the complexities of power sector planning and operation are modelled in the main MESSAGEix model, which provides new capacities to be installed to the Materials module. In which case, it would make sense to make reference to this instead of saying that the Materials module takes this into account.

Clarified as suggested.

L495 Some clarification would be useful here. Certain demands are exogenous (HVCs), certain are endogenously produced but by other parts of the larger model (oil for transport).

Beginning of section 2.4 explains some demands are exogenous and some are endogenous. Additions are made to clarify it better in Section 2.4 and 2.4.1.

L503 Since you mentioned materials flows and stocks several times in the introduction, I expected some more discussion on the methodological choices to calculate materials demand. By using a single equation of demand per capita, you are making the choice to model annual flows for the total material demand, while also detailing flows and stocks for one of the uses (power capacities). Some further justification of the methods used, discussion of pros and cons, and, possibly, anticipation of future work, would improve the manuscript. Also, please specify if the equations are for the total material demand or for total net of what is calculated endogenously.

Some more explanation about the justification of aggregate GDP method to project demand is added to Section 2.4.2. The advantages and disadvantages of endogenizing material demand vs using the aggregate GDP based demand are discussed more in the Discussion and Conclusion Section. Here we also mention the future work on endogenously covering material demands for example for buildings and vehicles.

Regarding the question about total vs net material demand: The material demand for aluminum, steel and cement are derived for the total of the material demand. The demand is exogenously projected for the total of HVC demand and only for the residual demands for methanol and ammonia, which are not covered by the endogenous representation as mentioned in Section 2.4.2.

L516 The method to derive chemicals demand is not justified. You derive income elasticities from other projections, when these projections themselves used some other methodology that was not described here. It would be more correct to derive income elasticities from historical statistics, or to re-use the same methodology as the source you refer to (IEA).

Unlike aluminum and steel, it is difficult to find a comprehensive historical data for chemicals therefore we had to opt for an alternative method to project the chemicals demand. The text is modified to describe the methodology that IEA uses. However, the source data that IEA use is not publicly available. Therefore, we use implicit income elasticities derived from their projections.

L559 When discussing Figure 9, only materials demand for the construction phase were discussed. These other flows for operation, maintenance and decommissioning were not discussed. This is missing.

Material intensities in Figure 9 (Figure 10 after revision) are used during construction phase and released at the end-of-life. Decommissioning is also mentioned in the explanation of Figure 9 (Figure 10 after revision) with these sentences: "Upon construction of power plants, a demand for the three bulk materials is generated endogenously based on the material intensities in Arvesen et al (2018) per generation technology, vintage and region. Power sector material stocks exhibit specific lifetimes, and upon the retirement of the capacity, end-of-life waste material is released which then is collected and becomes available for recycling". The operation and maintenance flows are not included in this model version. To clarify, this information is added to Section 2.4.1.

L567+ Some more discussion and critical review of the results would be welcome here. Why the differences in production levels with statistics, were not equations calibrated in order to reproduce these exact numbers? For steel, final energy is below statistics and emissions are above, while it's the other way round for chemicals, is there a reason for this? Final energy for aluminium and cement exactly matches statistics: is this because of rounding or is it by design?

Explanation related to the differences in production, final energy and CO2 emissions are added. Also the values are updated with the new model runs and updated statistics. The changes are:

- Aluminum, steel, chemical, cement production values are updated from the most recent model run.
- Aluminum statistics source changed to International Aluminum Institute 2020 global aluminum cycle.
- Steel production statistics changed to World Steel Association.
- Alumina refining stage added to the model. Therefore, the final energy and emissions value for aluminum are updated to the recent model run.
- Final Energy statistics value previously excluded the refining part for aluminum. It is updated to a value that includes refining as well.
- Final Energy and CO2 Emissions values for steel are updated to the most recent model run after the addition of new iron and steel technologies. In addition, the final energy value from the model for steel does not include the cokeoven inputs.
- The source of statistics for Final Energy steel is changed to IEA Energy Balances directly. From energy balances, what is reported under iron and steel and blast furnaces are summed and cokeoven sector is excluded.
- CO2 emissions statistics value for steel updated based on the new production value from World Steel Organization.
- CO2 Emissions for chemicals are updated to the value from the most recent model run.

L597 Your high value for hydro seems to stem from Arvesen et al 2018, and in this the high-range is due to the assumption of a plant in a remote location. Perhaps you can adopt a value closer to the majority of the literature. See for instance Chapter 12 in Ashby 2013: https://www.sciencedirect.com/book/9780123859716/materials-and-the-environment The studies referenced do specify which hydropower technology they assume, so you can make an informed choice on what technology or technology mix you want to represent. Arvesen et al 2018 is a reservoir and Kalt et al 2021 has a lower value and specifies that it is for a run-ofriver type. (Also, Deetman et al 2021 is not actually listed in the manuscript references, presumably it is this: https://doi.org/10.1016/j.resconrec.2020.105200)

Arversen et al., 2018 provides three different material intensities for the hydropower plant. "1" is a hydropower plant with reservoir in a remote location, "2" is a hydropower plant with reservoir in nearer location and "mix" is both types considered by using market shares of these types. In our model we represent both run-of-river and reservoir types but not exceptional cases such as reservoirs that are in remote locations. Therefore, we choose to use Kalt et al., 2021 medium value for hydropower to be closer to the literature average and as it better fits to the types of hydropower we represent in our model.

The reference for Deetman is included in the list of references:
Deetman, S., Boer, H. S. de, van Engelenburg, M., van der Voet, E., and van Vuuren, D. P.: Projected material requirements for the global electricity infrastructure – generation, transmission, and storage, Resources, Conservation and Recycling, 164, 105200, DOI: 10.1016/j.resconrec.2020.105200, 2021.

L615 Does your 2 degrees scenario need to reach net-zero CO2 emissions? What is the carbon budget assumption? This sounds rather like a 1.5C scenario.

2 degrees scenario can reach net-zero and go below to net-negative emissions globally to bring back cumulative CO2 emissions to within the budget by 2100. But there is not a net-zero constraint to force the model. Instead, a carbon budget assumption of 1000 GtCO2 is implemented. This is also added to the text. As a result of the implemented budget, model reaches net zero between 2070 – 2080 and has negative emissions in 2080.

L627 "may stem": You should be able to explain your modelled findings with certainty.
Changed as suggested.

L637 & Figure 12: The different level in 2020 is surprising to me. By doing modelling enhancements, the model has deviated further from statistics? I would have expected that emissions not covered in the materials sub-sectors to be reported in the other industry sector, in order to have total industry emissions coincide with statistics. Also, given the large difference in 2020 emissions, it doesn't make much sense to compare cumulative emissions unless you harmonize the two series to a common starting level.

We have done some further calibration of 2020 industry emissions in the new model version and now it is closer to the 9000 which reported by EDGAR6 and CEDS data sources. Also, the model version without materials is updated to a more recent calibration. This reduced the emission difference between two models. Both materials and non-materials version cover the whole industry. There is still a small amount of difference remaining between the two models as the calibration methods used are different for these two model versions as a result of the structural differences that comes with the materials module. However, this difference is within the uncertainty limits (5%) of the reported industrial emissions. The comparison related to cumulative emissions is removed and the explanation in the section is updated.

L642 From my understanding of earlier sections, the model does not account for the price elasticity in materials demand, i.e. demand is entirely inelastic, whereas you mention here that price inelasticity is not accounted for.

It should have been price elasticity. Changed.

L651 Some introduction is needed here, to transition from the version comparison earlier to the discussion of results within the new version only.

Changed as suggested.
L661 This paragraph would benefit from some discussion on what drives the relative flexibility of sectors' emissions.

More discussion is added as suggested.

L666 For aluminium, a mention of the indirect emissions of power production would flesh out the findings.

Added as suggested.

L669 As both materials industries and other sector are bundled together in Figure 13 panels b and c, it is impossible to disentangle the effects of the new modelling from what is happening in the other sector. Some results are likely largely driven from other sector (like the increase in electrification). Sectoral results could be included in the supplementary material.

This section is to get an overview of the total industry final energy. Graphs for specific industry sectors are provided and discussed more in detail in the rest of the results section. In addition, panel c non-energy use does not have any other sector but completely the newly added model results. To provide more information on the other sector final energy, a graph is added to the supplementary material (Figure S6).

L679 The description of the situation in 2020 calls for a sentence on how this changes in the projected years.

Added as suggested.

L708 As mentioned, not including CCS or hydrogen pathways for steel significantly reduces the mitigation potential of this sector, and the relevance of these findings.

Those technologies are added to the new model version as suggested.

L713 Where is the additional scrap coming from in the 2C scenario? Does the model use all scrap that is available or is there some scrap that is not used because it is uneconomic?

In No Policy scenario not all the scrap is used as not all regions have the electric arc furnace capacity and it is not economic to install these from zero just to use the scrap. However, in 2 degrees scenario, there is an increased adoption of electric arc furnaces motivated by reducing the emissions and as a result more scrap is used. The total scrap availability between two scenarios do not change except the minor effect of power sector scrap that is endogenously released that can be different in scenarios. This information is added to the text as well.

L722 Why does CCS only emerge after 2050 for cement but is an option from 2030 for ammonia and methanol? In general, the findings of much more $CO_2$ captured from petrochemicals production compared to $CO_2$ captured from cement production is surprising.

Is this a finding driven by techno-economic parameters in your modelling or due to different sectoral availability assumptions for CCS?

This question is answered in the first section of the response and more information is added to the cement results section.

L726 Figure 16: the increase in electricity share is considerable, what is driving this? Do you model an electric kiln option?

The increase comes both from electric kiln and CCS electricity requirements. This is added to the text.

L729 As supply is driven by demand, consider switching the order of the sections by putting demand first.

The order is changed as suggested.

L748 This finding is quite important, as it rebuts the story that the decarbonization transition will have a rebound effect that will result in more emissions.

L802, L825 No need to mention "will" here.

Removed as suggested.

L845 What biophysical limitations on material demand are you envisaging to consider?

Biophysical limitation mentioned here refers to the improvements envsiaged to cover a broader range of material stocks and flows that can reflect material demand better instead of GDP based projections. Material flows directly linked to service demand (e.g., passenger kilometer mobility demand) will offer a more accurate estimation of resource needs, while accounting for stocks based on their lifetime will improve the estimation of end-of-life flows and the availability of recyclable materials. The sentence is rephrased.

**Referee #2:**

This manuscript offers a detailed description of the MESSAGEix-Materials model as well as an overview of some of its results. While it acts mostly as documentation for the model (which is suspect is OK for this journal), it is an interesting, and a timely contribution to the IAM literature, showing the further development of IAM modelling beyond the traditional focus of land, energy, and emissions, broadening into material stocks and flows.
I don't have any fundamental issues with the manuscript or the methodology, but I do think the manuscript can be improved to make clearer to the reader how many of the components were implemented, and also clarify what the methodological advance is, its purpose, and its potential future use. Below I give some examples of areas which the authors may want to focus on.

Figure 1 is hugely useful, however also a bit confusing. According to the figure, Material Demand is completely exogenous – which seems to be in contradiction with both the motivation of the paper, as well as results shown later. Similarly, "Energy Demand" is shown as exogenous, which is definitely not the case for MESSAGE. If the figure only focuses on the

Materials module of MESSAGEix-Materials, this should be made clear in the labels and caption of the figure.

In various parts of the manuscript, with the revised version we try to make it clearer to the reader what is included in the model and what is not yet included. Information on the methodological advance, its purpose and potential future uses is distributed over introduction and discussion and conclusion sections. We improve the discussion and conclusion section to provide some more concrete examples on the future uses and to emphasize the methodological advance and its purposes.

Regarding the comments about Figure 1 (Figure 2 after revision), material demand is not completely endogenized yet, but this is the ultimate goal. To clarify this in the figure (that not all of material demand is exogenous), we change the label as "Rest of the material demand". Energy demand in the figure refers to the useful energy demand which is exogenous in MESSAGE (https://docs.messageix.org/projects/global/en/latest/energy/demand.html). Fuels used to satisfy the useful demand and therefore final energy is determined in the model. Energy demand in the figure is the useful energy that is provided exogenously for the "other industry", residential and commercial and transport sectors. The label is changed as Rest of Useful Energy demand. The dashed area is the system boundary of MESSAGEix-Materials and outside of the dashed area is not covered endogenously in the model which is shown in the figure legend. Some clarifications are added to the text about what is included and what is not.

In a number of cases Trade is mentioned, and I commend the authors for including this very important component. Can a few more details be given for how trade is modelled and calibrated? How are supply regions determined, and what dynamics do future projections of trade follow? How are costs and supply potentials determined?

As suggested, more details are added on how trade is modeled, calibrated and what type of costs are considered. Also, information is added on how supply regions and potentials are determined.

On line 348 the lack of trade for cement is justified. However, I think this statement should be clearer and the decision better justified. I understand and agree with the authors have made here, but the reasoning may not be clear to readers who have not dealt with modelling trade across large aggregation levels.

We added more justification about the lack of trade representation in the model for cement.

On page 30 the different technologies for the steel industry and cement industries are shown. Is it possible to compare these results with other existing similar exercises. Specifically, van Sluisveld et al. 2021 and Muller et al. 2024.

A comparison of results for iron&steel and cement industries with the similar studies is added in Section 3.3.2, Iron and Steel Industry such as: van Sluisveld et al. 2021, Muller et al. 2024 and Keremidas 2024.

The manuscript would benefit a lot from a further elaboration on what these new modelling capabilities offer. For instance it is mentioned that this model can improve the modelling of the "circular economy" (line 824) or "decent living standards" (line 845), but the explanations are very vague and cursory. It would be useful if this discussion is expanded a bit. Generally the

discussion section would benefit from better structuring, and perhaps the use of sub-sections to help guide the reader. Currently it seems like a lot of (valid) after-thoughts compiled together.

More details are added related to circular economy and decent living standards applications and the benefits that the new model advancements bring. In addition, the discussion and conclusion section is divided into 3 sections for a better structuring.

Minor comments
- L39: Not all PE IAMs are optimization models. In fact the IMAGE model which is cited in the following sentences is recursive-dynamic
  Removed the word optimization.
- L193: It is mentioned that there is a maximum recycling rate constraint. The parameterisation used should be provided in the supplementary information. The parameterisation and formulation used for recycling is added to the supplementary information.
- L313: The term "significant" is vague. It is better to mention absolute values or how they compare with other technologies. The term "significant is removed and absolute values are added.
- A general comment about many of the figures: Some of the figures showing the reference systems are hard to read, would be better suited in landscape format. For the figures that include more information such as aluminum and steel, or generic reference material system, the figures are converted to landscape format.
- L825 an article is needed before "circular economy". An article before "circular economy" is added.